# Empiric azithromycin alters the upper respiratory microbiome and resistome without anti-inflammatory benefit in COVID-19

Azithromycin is a widely used antibiotic and was frequently used to treat hospitalized patients during the COVID-19 pandemic. The impact of empiric azithromycin use on the respiratory microbiome in patients with viral respiratory infections is unclear. Here we used longitudinal metatranscriptomics on nasal swabs from a prospective multicentre cohort of 1,164 patients hospitalized for COVID-19. We compared the upper respiratory microbiome, resistome and systemic immune response in patients treated with azithromycin ($n$ = 366) with those who received no antibiotics ($n$ = 474) or other antibiotics ($n$ = 324). We found that azithromycin altered microbiome composition and increased the expression and relative proportion of macrolide/lincosamide/ streptogramin (MLS) resistance genes. These changes occurred after 1 day of exposure and persisted for over a week. MLS resistance gene expression was associated with commensals and potential pathogens, while there were no differences in host inflammatory gene expression in blood and airways. This demonstrates that empiric azithromycin treatment impacts the upper respiratory microbiome and resistome without apparent anti-inflammatory benefit.

Azithromycin, a World Health Organization essential medicine[1], is one of the most widely used antibiotics in human healthcare with >40 million prescriptions annually in the United States alone[2]. Azithromycin overuse has been well documented in the outpatient setting[3], where an estimated 30% of antibiotic prescriptions are inappropriate[4]. During the first year of the coronavirus disease 2019 (COVID-19) pandemic, azithromycin became one of the most commonly used antibiotics in hospitalized patients as well[5,6]. This was driven in part by early studies suggesting possible antiviral activity[7,8] and previous work demonstrating anti-inflammatory properties of macrolide antibiotics[9,10].

Randomized controlled trials, however, subsequently demonstrated that azithromycin conferred no clinical benefit in the treatment of COVID-19 (refs. 11–15). These included hospitalized patients (RECOVERY trial[13], $n$ = 7,763), hospitalized patients with severe COVID-19 (COALITION II trial[14], $n$ = 397), mild–moderate COVID-19 (ATOMIC-II[12], $n$ = 298) and outpatients (PRINCIPLE[15], $n$ = 1,323). Nonetheless, many medical centres initially incorporated azithromycin into their COVID-19 treatment guidelines, and public misinformation continues to drive overprescription of the drug[16].

Recent work has found that azithromycin exposure can alter the human microbiome and its reservoir of antimicrobial resistance genes, collectively termed the resistome[17–19]. For instance, secondary analyses of the Macrolide Oraux pour Réduire les Décès avec un Oeil sur la Résistance (MORDOR) clinical trials found that biannual mass azithromycin distribution to African children led to an increase in the abundance of both macrolide and other antimicrobial resistance gene (ARG) classes in the gut microbiome[17]. In addition, adults with asthma randomized to thrice weekly azithromycin over 12 months had an increase in PCR copy number of macrolide resistance genes in sputum samples compared to controls[20].

Despite being the most common scenario for its use, no studies have yet assessed the impact of azithromycin on the respiratory

✉e-mail: chaz.langelier@ucsf.edu

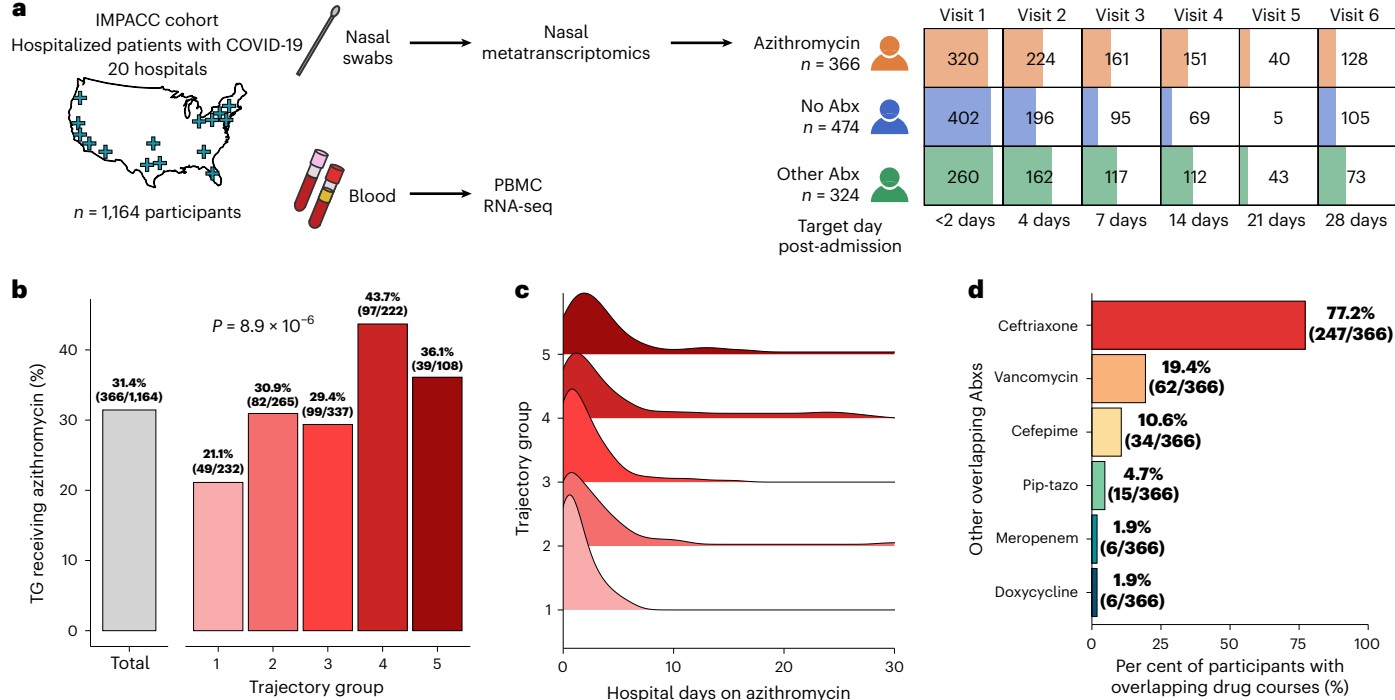

**Fig. 1 | Study overview and cohort demographics. a**, Graphical overview showing the geographic location of the IMPACC cohort study sites, sampling approach and antimicrobial exposure groups studied, including those ($n$ = 366, 31.4%) treated empirically with azithromycin ± other antibiotics, those ($n$ = 474, 40.7%) who received no antibiotics and those ($n$ = 324, 27.8%) who received antibiotics other than azithromycin. **b**, Bar plot showing the per cent of patients within each COVID-19 trajectory group (colours) exposed to azithromycin. Grey reference bar indicates per cent of patients within the cohort treated with azithromycin. Significance calculated using Kruskal–Wallis test. **c**, Density plot highlighting distribution of azithromycin treatment with respect to days of hospitalization. **d**, Bar plot depicting antimicrobials most frequently co-prescribed with azithromycin. Pip-tazo, piperacillin-tazobactam.

microbiome in the context of empiric prescription for acute respiratory infection. Furthermore, no studies of azithromycin exposure have yet incorporated metatranscriptomics, which can assess both bacterial 16S rRNA abundance and ARG expression, providing a functional profile of the actively expressed resistome[21,22].

To address these gaps, we carry out respiratory metatranscriptomics in a prospective cohort of 1,164 adults hospitalized for COVID-19 and study the impacts of azithromycin exposure. We find marked changes in the respiratory microbiome, including increases in detectably expressed macrolide resistance genes and their proportional representation in the airway resistome, without evidence of antiviral or immune-modulating benefit. Taken together, our findings offer insights into the adverse effects and biological consequences of empiric azithromycin exposure during viral infection.

## Results

### Cohort

We carried out a prospective observational study of 1,164 adults hospitalized for COVID-19 enrolled in the multicentre IMmuno Phenotyping Assessment in a COVID-19 Cohort (IMPACC)[23–25] between May 2020 and March 2021 (Fig. 1a). Previously established COVID-19 outcome trajectory groups (TGs)[24] were utilized to group patients on the basis of disease severity. TGs ranged from 1 (lowest severity) to 5 (death within 28 days)[24]. Administration of azithromycin and other antibiotics was tracked following admission and throughout hospitalization. Of 1,164 patients with COVID-19 studied, 366 (31.4%) were treated empirically with azithromycin ± other antibiotics (Azithro group), 474 (40.7%) received no antibiotics (No-Abx group) and 324 (27.8%) received antibiotics other than azithromycin (Other-Abx group) (Extended Data Tables 1 and 2 and Extended Data Fig. 1).

Empiric azithromycin administration was most common among patients with the highest COVID-19 severity (Fig. 1b), although

compared with those in the Other-Abx group, azithromycin-treated patients had less severe disease (Extended Data Table 1). The median number of azithromycin treatment days was 2 (interquartile range (IQR) 1–4 days, range 1–35 days), which significantly differed across TGs ($P$ = 0.03, Extended Data Fig. 2). Azithromycin was administered in most (98.2%) patients within 1 week of hospital admission (Fig. 1c). Patients treated with azithromycin were most likely to have been co-administered ceftriaxone (77.2%) or vancomycin (19.4%) (Fig. 1d). Azithromycin usage did not differ based on sex or race (Extended Data Table 1).

### Azithromycin exposure alters the respiratory microbiome

We first examined the impact of azithromycin on the upper respiratory tract microbiome using metatranscriptomic RNA sequencing (RNA-seq) of nasal swab (NS) samples collected at 6 timepoints over 28 days following hospital admission. These analyses were adjusted for age quintile, sex, severity TG, days of hospitalization, patient, receipt of corticosteroids, and receipt of the six most common antibiotics aside from azithromycin. We found that azithromycin treatment for 5 ± 1 days, a common duration of prescription[26], was associated with a significant decrease in bacterial relative abundance in the airway (adjusted $P$ value ($P_{adj}$) = 0.026, fold change (FC) = 0.82), with an effect observable within 1 ± 1 days ($P_{adj}$ = 0.0019, FC = 0.90; Fig. 2a). Other antibiotics also led to a decrease in upper respiratory bacterial relative abundance after 1 ± 1 days ($P_{adj}$ = 0.036, FC = 0.93) but not at the later timepoint (Fig. 2a).

Assessment of the mycobiome demonstrated that receipt of azithromycin was associated with an increase in fungal relative abundance in the upper airway after 1 ± 1 days ($P_{adj}$ = 0.038, FC = 1.17; Fig. 2b), with a time-dependent increasing trend observed over 5 days of azithromycin administration (Extended Data Fig. 3). No differences in upper airway microbiome alpha diversity were observed based on

azithromycin treatment status (Fig. 2c). However, significant differences were found in microbiome community composition based on azithromycin exposure, measured by the Bray–Curtis dissimilarity index (permutational multivariate analysis of variance (PERMANOVA) $P = 0.001$, Fig. 2d). A comparison of the trajectories for Bray–Curtis distances versus the earliest timepoint for each patient demonstrated marked shifts in community composition over time, independent of antibiotic exposure (Fig. 2e).

Differential taxonomic abundance analysis demonstrated that azithromycin exposure was associated with enrichment of potentially pathogenic taxa in the upper airway including *Staphylococcus* and *Klebsiella* species, and depletion of several typically commensal taxa such as *Neisseria* and *Fusobacterium* (Fig. 2f). Within differentially abundant genera, we also examined the relative abundance of the most prevalent species in the upper airway microbiome (Fig. 2g). For example, we found that *S. aureus* and *S. epidermidis* accounted for the majority of differentially abundant *Staphylococcus* spp. based on azithromycin exposure.

We next performed a functional analysis of microbial metabolic pathway expression using HUMANn3[27] and MaAsLin2 (ref. 28) to compare the Azithro group to the Other-Abx and No-Abx groups at the $5 \pm 1$ day timepoint, but found zero differentially abundant pathways after adjusting for multiple comparisons. We also tested whether azithromycin treatment was associated with any changes in SARS-CoV-2 relative abundance in the upper airway. We observed no differences with respect to either the Other-Abx or No-Abx groups after $5 \pm 1$ days of treatment (Extended Data Fig. 4).

## Azithromycin exposure alters the respiratory antimicrobial resistome

To understand the impact of azithromycin exposure on the respiratory antimicrobial resistome, we first compared ARG alpha diversity across groups. Azithromycin exposure was associated with an increase in the resistome Shannon diversity index at the early ($1 \pm 1$ day) treatment timepoint compared to the Other-Abx group ($P_{adj} = 5.8 \times 10^{-3}$, FC = 1.24) but not to the No-Abx control group (Fig. 3a). A shift in the composition of the antimicrobial resistome upon azithromycin exposure was also evident, with significant differences in Bray–Curtis dissimilarity indices observed at $1 \pm 1$ days and $5 \pm 1$ days of azithromycin treatment versus controls (PERMANOVA $P = 0.001$; Fig. 3b).

Since azithromycin is a macrolide antibiotic, we next focused on ARGs conferring resistance to the MLS class of antibiotics. We found that azithromycin exposure was associated with a significantly greater number of detectably expressed MLS genes compared with the No-Abx or Other-Abx groups at both the early ($P_{adj} = 3.5 \times 10^{-4}$ and $6.2 \times 10^{-7}$, FC = 1.48 and 1.61, respectively) and late ($P_{adj} = 1.3 \times 10^{-3}$ and $4.3 \times 10^{-6}$, FC = 1.61 and 1.70, respectively) timepoints (Fig. 3c). Longitudinal modelling subsequently demonstrated that days of azithromycin exposure resulted in a significant increase in the number of

detectably expressed MLS ARGs in the upper respiratory microbiome ($P_{adj} = 7.6 \times 10^{-7}$, Fig. 3d). These findings were robust to the variation in disease severity between groups as demonstrated by a sensitivity analysis stratifying the model by disease severity (less severe, TG 1–3; more severe, TG 4–5) (Extended Data Fig. 6).

To comparatively evaluate the impact of azithromycin exposure across different ARG classes, we assessed longitudinal changes in the fraction of the resistome represented by each class. We found that after $5 \pm 1$ days of azithromycin exposure, MLS ARG proportion increased from 24.5% to 42.9% of the resistome ($P_{adj} = 1.7 \times 10^{-4}$, Fig. 3e). Notably, the enrichment of MLS ARGs, both in terms of ARG richness and as a proportion of the resistome, persisted even 7–10 days after cessation of azithromycin (Fig. 3f and Extended Data Fig. 5).

## Correlations within the airway resistome and microbiome link taxa to resistance genes

Multiple MLS resistance genes were identified within the airway microbiome, with *ermA* and *msrA* being most prevalent (Fig. 4a). To assess relationships between MLS resistance genes and bacterial taxa within the upper airway microbiome, we performed multidimensional correlation analyses (Fig. 4b). Significant positive correlations were found between several MLS genes and both potentially pathogenic taxa (for example, *Staphylococcus, Streptococcus)* as well as common commensals (for example, *Corynebacterium*[29], *Dolosigranulum*[30]) (Fig. 4b,c and Supplementary Table 1). Relationships were further reinforced with species-level multidimensional correlation analyses (for example, *Staphylococcus aureus* and *mphC, msrA*) (Extended Data Fig. 7 and Supplementary Table 2). Application of the Comprehensive Antibiotic Resistance Database (CARD) Resistance Gene Identifier (RGI) pathogen-of-origin tool[31], which matches *k*-mers from detected ARG sequences against a database of established bacterial pathogens, provided additional insight regarding linkages between macrolide resistance genes and specific species (Extended Data Fig. 8). For instance, this additional analysis highlighted a relationship between *Streptococcus pyogenes* and *ermA*, and additional linkages between *ermC* and both *Staphylococcus aureus* and *Staphylococcus epidermidis*.

## Azithromycin does not change host inflammatory responses

Previous studies have found that azithromycin can confer anti-inflammatory properties, leading to off-label use of the antibiotic as an immune modulatory agent[9,10]. We thus sought to understand whether azithromycin exposure in the setting of acute COVID-19 was associated with changes in host inflammatory gene expression in the airway or blood by carrying out differential gene expression analyses. No differentially expressed genes were identified (FDR < 0.05) in either the airway or blood after $5 \pm 1$ days of treatment (Supplementary Tables 3 and 4), suggesting that azithromycin does not meaningfully attenuate pathologic inflammatory responses in hospitalized patients with COVID-19.

**Fig. 2 | Azithromycin treatment alters the respiratory microbiome. a**, Total bacterial relative abundance in the nasal microbiome, measured by RPM, comparing patients treated with azithromycin (orange), other antibiotics (green) or no antibiotics (blue) after $1 \pm 1$ days (left) or $5 \pm 1$ days (right) of antimicrobial treatment or hospitalization (controls). **b**, Total fungal abundance in the nasal microbiome, measured by RPM, highlighting differences between antimicrobial treatment groups. **c**, Alpha diversity of the nasal microbiome highlighting differences between antimicrobial treatment groups. Significance in **a**–**c** calculated using a pairwise two-sided Wilcoxon rank-sum test with Benjamini–Hochberg correction. Sample sizes for groups in **a**–**c** are: No Abx $1 \pm 1$ days, $n = 666$; No Abx $5 \pm 1$ days, $n = 120$; Azithro $1 \pm 1$ days, $n = 394$; Azithro $5 \pm 1$ days, $n = 116$; Other Abx $1 \pm 1$ days, $n = 396$; and Other Abx $5 \pm 1$ days, $n = 99$. **d**, PCoA of Bray–Curtis dissimilarity reveals compositional differences of the nasal microbiome based on azithromycin treatment for $1 \pm 1$ days (orange) or $5 \pm 1$ days (red) in comparison to no antibiotic exposure (blue). Significance calculated using PERMANOVA. $R^2$ and $F$ statistics are provided in the Source Data

file. **e**, Bray–Curtis dissimilarity distances over time within the nasal microbiome compared with the earliest date of sampling following hospital admission. Shaded interval denotes 95% confidence interval, calculated with a GAMM. **f**, Differentially abundant microbial genera in patients treated with azithromycin for $5 \pm 1$ days ($n = 116$) compared to other patients (No-Abx + Other-Abx groups, $n = 219$). log fold change (lFC) values are plotted on the *x* axis. Taxa enriched with azithromycin treatment are in orange; those depleted are in grey. All adjusted $P$ values are $q < 1 \times 10^{-9}$. **g**, Species-level proportional representation in the upper airway microbiome for the most differentially abundant taxa (by lFC) from **f**. For each genus, the top two most prevalent species are plotted, with additional species aggregated as 'Other spp.'. Azithro group is in orange ($n = 116$); patients who were not treated with azithromycin (No-Abx + Other-Abx groups) are in grey ($n = 219$). Boxplot limits correspond to the IQR and the centre line denotes the median. The lower whisker extends to the smallest value within 1.5× IQR below Q1, and the upper whisker extends to the largest value within 1.5× IQR above Q3.

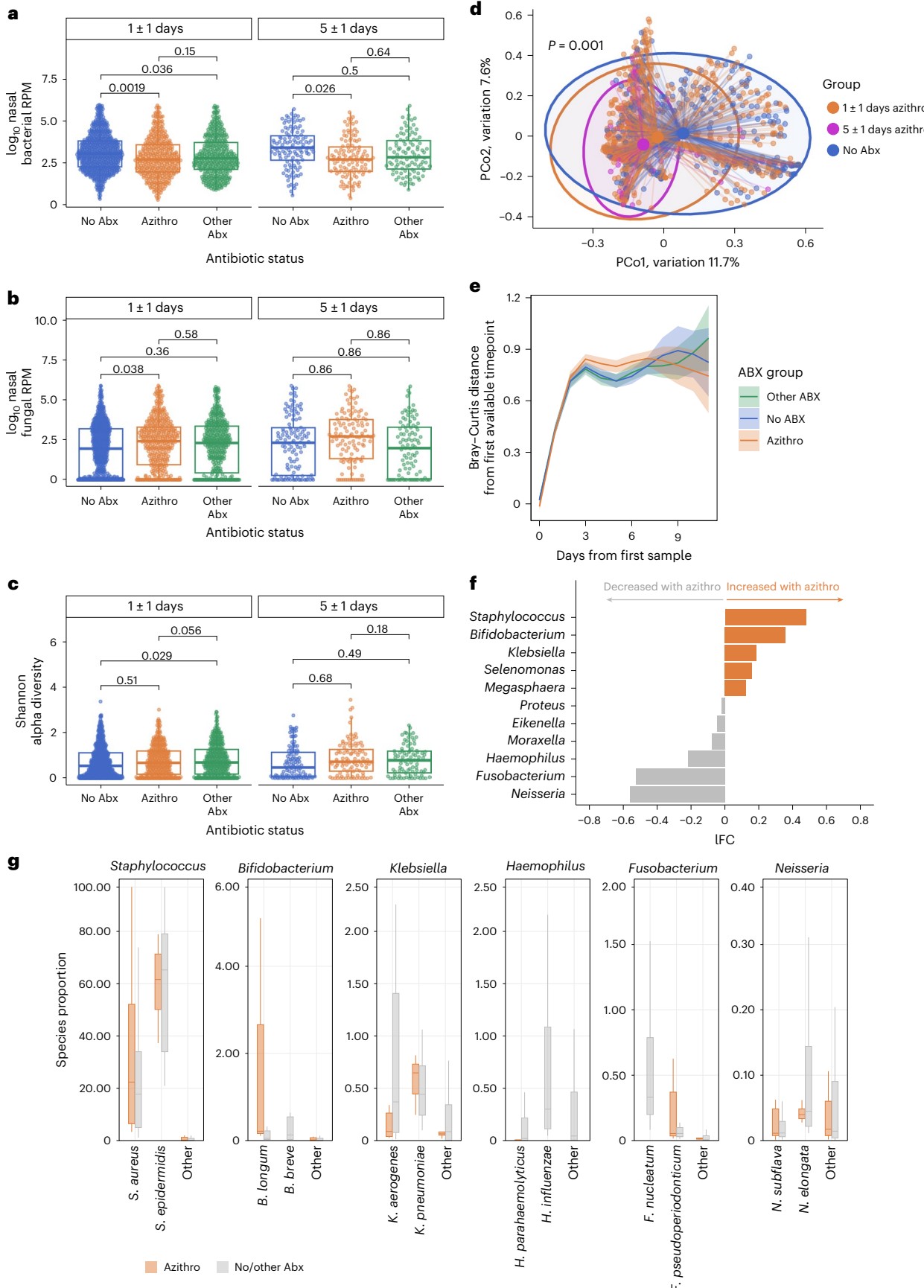

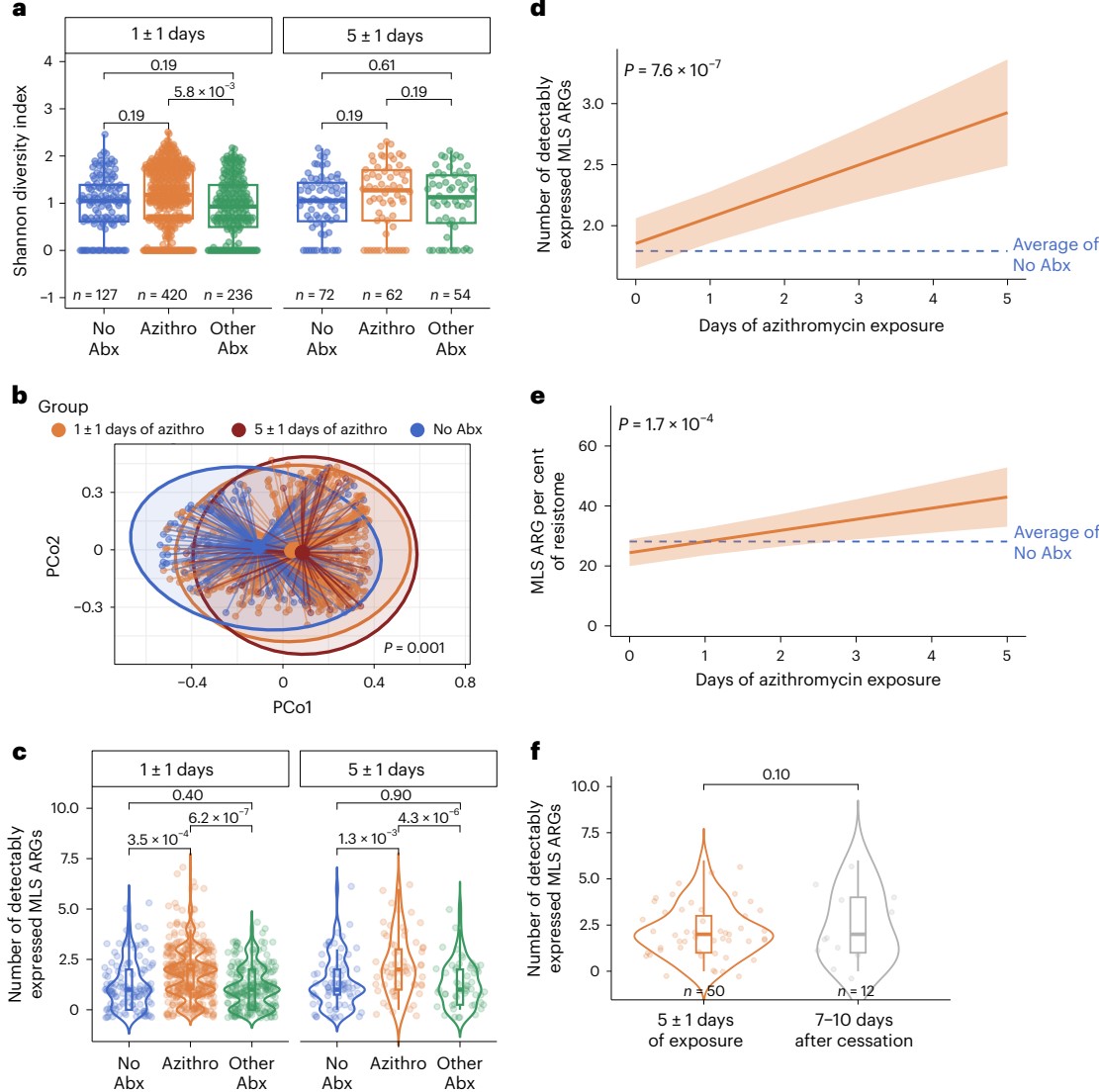

**Fig. 3 | Azithromycin exposure alters the respiratory resistome. a**, Alpha diversity of the nasal antimicrobial resistome highlighting differences between patients treated with azithromycin (orange), other antibiotics (green) or no antibiotics (blue) at 1 ± 1 days (left) or 5 ± 1 days (right) of antimicrobial treatment or hospitalization (controls). *P* values were generated using a linear mixed-effects model adjusted for age quintile, sex, severity TG, days from hospitalization and receipt of steroids as fixed effects in the model, and the participant's enrolment site as a random effect. For the only-Azithromycin group, the number of days of the six most common co-administered antibiotics (vancomycin, ceftriaxone, cefepime, piperacillin-tazobactam, doxycycline and meropenem) was also included as a fixed effect. Resulting *P* values were adjusted using the Benjamini–Hochberg FDR algorithm. **b**, Compositional differences of the nasal resistome based on azithromycin treatment for 1 ± 1 days (orange, *n* = 420), azithromycin treatment for 5 ± 1 days (dark orange, *n* = 62) or no antibiotic treatment (blue, *n* = 199). *P* value generated using a PERMANOVA test. $R^2$ and $F$ statistics are provided in the Source Data file. **c**, Number of detectably expressed MLS resistance genes (ARGs) based on antibiotic treatment groups. Group *n* values are identical to those in **a**. *P* values shown were generated using the same

linear mixed-effects model and corrected using the Benjamini–Hochberg FDR algorithm. **d**, GAMM showing changes in the number of detectably expressed MLS resistance genes over time. **e**, GAMM demonstrating longitudinal changes in the proportional representation of MLS resistance genes (orange) in the nasal resistome over time. **f**, Number of detectably expressed MLS resistance genes after 5 ± 1 days of exposure compared with 7–10 days after azithromycin cessation. Significance was calculated using a two-sided Wilcoxon rank-sum test. For boxplots, the box limits correspond to the IQR and the centre line denotes the median. The lower whisker extends to the smallest value within 1.5× IQR below Q1, and the upper whisker extends to the largest value within 1.5× IQR above Q3. For violin plots, the shape of the violin represents the kernel density estimate of the data, with tails trimmed to the upper and lower ranges of the data. For GAMM plots, a smoothed curve representing the estimated nonlinear relationship between variables is plotted. The centre line represents the predicted mean value, and the shaded area denotes the 95% confidence interval. Significance of the GAMM terms was assessed using likelihood-ratio tests (ANOVA), comparing nested models with and without the terms of interest.

## Discussion

In a large multicentre cohort of hospitalized patients with COVID-19, empiric azithromycin treatment was associated with changes in the upper respiratory tract microbiome, mycobiome and antimicrobial resistome. We observed a significant expansion of detectably expressed macrolide resistance genes after 5 ± 1 days of azithromycin treatment,

with effects in some cases observed within a few days. In addition, we found that azithromycin treatment was associated with changes in the composition of the upper airway microbiota including enrichment in potentially pathogenic taxa such as *Klebsiella* and *Staphylococcus* species. Together, our findings demonstrate that inappropriate azithromycin use in patients with viral respiratory infections can drive expansion

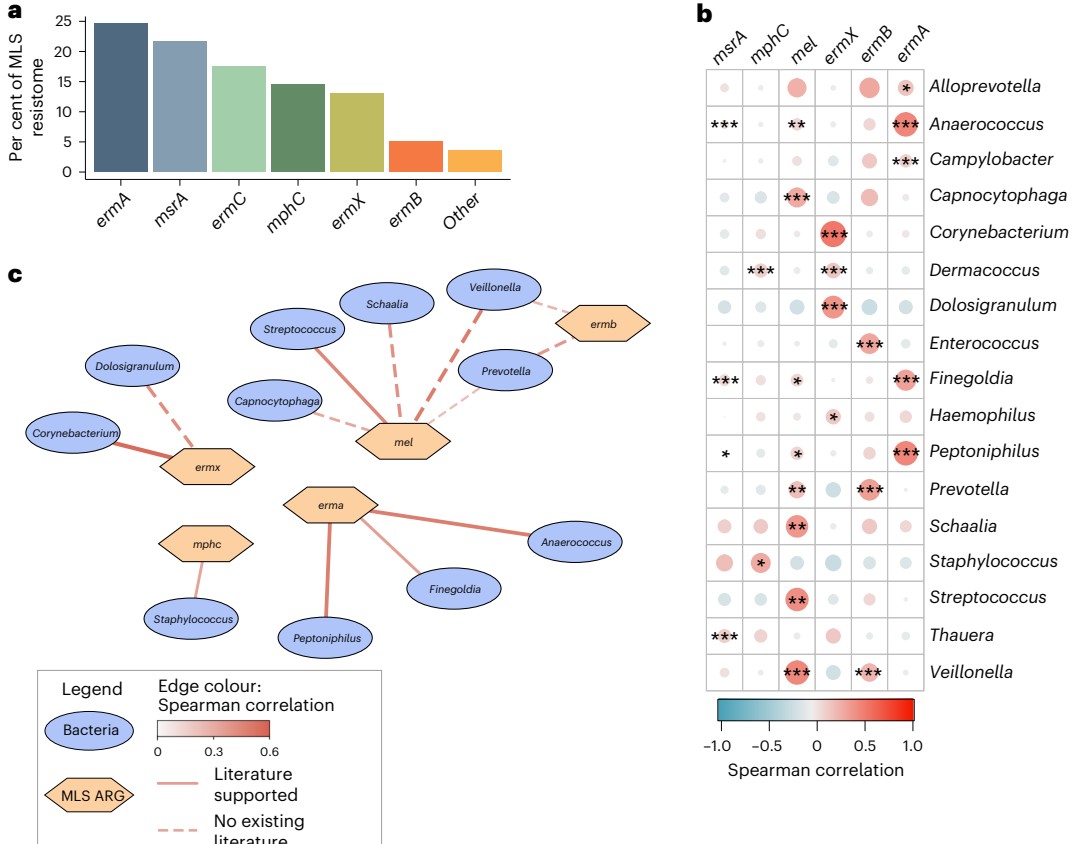

**Fig. 4 | Correlations within the airway resistome and microbiome. a**, Per cent of the MLS resistome across the full cohort comprised by individual MLS ARGs. **b**, Correlations between relative abundance of MLS resistance genes and the most abundant bacterial genera across the cohort. Among the 30 most abundant genera, those with significant associations are displayed. Colour bar reflects Spearman correlation coefficient. *P* values were generated for all pairwise comparisons using a two-sided Spearman correlation test and corrected for multiple testing using the Benjamini–Hochberg FDR algorithm. ***$P_{adj}$ < 0.001; **$P_{adj}$ < 0.01; *$P_{adj}$ < 0.05. **c**, Network plot demonstrating significant correlations (Rho > 0.2 and $P_{adj}$ < 0.05) between bacterial taxa and MLS resistance genes. Taxa depicted by blue ovals. Resistance genes depicted by orange hexagons. Weight of edges and colour represent Spearman correlation coefficient. Solid lines show relationships supported by literature. Dashed lines show relationships where no association has previously been described in the literature.

of macrolide resistance determinants and disrupt the composition of the airway microbiome.

Previous work examining mass azithromycin treatment in African children[17,19] found a concerning relationship between exposure to this drug and an increase in macrolide resistance genes in the gut microbiome. We build on these important findings by demonstrating effects in the respiratory tract and at the transcriptional level, detectable within a few days of antibiotic treatment. Importantly, we find that azithromycin exposure correlates not only with an increase in the potential for resistance within the microbiome, but also with the functional expression of MLS resistance genes.

Macrolide-resistant *Streptococcus pneumoniae* and *Streptococcus pyogenes* are considered urgent threats by the US Centers for Disease Control and Prevention[32]. Perhaps it is not surprising that macrolide resistance in these species has increased over the past decade given our results and considering that 30% of antibiotics prescribed in outpatient settings have been deemed inappropriate or unnecessary[4]. Given that a large fraction of azithromycin prescriptions are written for children[3], this is particularly concerning, as they may become colonized with macrolide-resistant bacteria at an early age due to unnecessary exposure to this drug.

Azithromycin is used prophylactically for chronic obstructive pulmonary disease[33], cystic fibrosis/bronchiectasis[34], lung transplantation[35], HIV/AIDS[36] and other conditions. While few studies have examined the impact of azithromycin prophylaxis on the respiratory resistome, a recent study of asthma patients found an increase in macrolide resistance genes using multiplex PCR in the sputum microbiome after 12 months[20]. Further work is needed to understand the impact of azithromycin prophylaxis on the upper and lower respiratory microbiome and resistome in these patient populations.

A subanalysis of the MORDOR trial found that 4 years of biannual azithromycin treatment in African children reduced mortality but led to an increase in macrolide-resistant *Streptococcus pneumoniae* cultured from the nasopharynx[19]. Consistent with these previous microbiological observations, our correlation analyses suggested that both *Streptococcus* and *Staphylococcus* species, encompassing some of the most important bacterial pneumonia pathogens (for example, *S. aureus* and *S. pyogenes*), may harbour these resistance determinants. In addition, we found relationships between MLS resistance gene expression and the abundance of commensal and contextually pathogenic taxa, such as *Corynebacterium* species.

In the MORDOR trial, mass azithromycin treatment was also found to cause an increase in the burden of non-macrolide resistance genes in the gut microbiome[17]. In contrast, we found relatively few off-target effects of azithromycin on other classes of ARGs in the upper airway. One possible explanation may lie in the age and demographic differences of the studied populations. Alternatively, differences may be attributable to sampling of the respiratory versus gut microbiome, or the use of metatranscriptomics versus metagenomic DNA sequencing.

Previous work has demonstrated that azithromycin has immune-modulating potential[9,10], findings that have encouraged its use in patients with cystic fibrosis, chronic obstructive pulmonary disease[37] and other inflammatory diseases. Azithromycin use early during the COVID-19 pandemic was driven in part by the idea that it might attenuate harmful inflammatory responses; however, clinical trials eventually found no therapeutic benefit[11–15]. Consistent with this, we observed no significant associations between azithromycin treatment and inflammatory gene expression, or viral load, in either the airway or blood of hospitalized patients with COVID-19.

Strengths of our study include a large multicentre cohort, detailed clinical phenotyping, use of respiratory metatranscriptomics, and rigorous quality control of clinical and biological data. As with any research, our study also has limitations. These include the observational study design and the use of short read sequencing, which precluded definitively linking macrolide resistance genes to specific taxa. In addition, our analyses of nasal swabs, a widely available specimen type from patients hospitalized for COVID-19, were limited to the upper airway and thus may not reflect microbial changes occurring in the lungs.

Antibiotic administration data were extracted manually by clinical research coordinators at each study site, an approach susceptible to human error. Azithromycin treatment, however, was confirmed by an independent adjudicator for every patient. Because our study preceded widespread COVID-19 vaccination, results were not influenced by vaccination status but could potentially differ in a contemporary vaccinated cohort. Future randomized clinical trials that ideally include airway and gut microbiome sampling as well as bacterial culture are needed to more fully characterize the impact of exposure to azithromycin and other antibiotics on the human microbiome and resistome. Evaluation of lower airway samples and metagenome-assembled genomes could further extend our understanding of antibiotic exposure, including on specific taxa carrying ARGs.

In sum, we find that azithromycin exposure in hospitalized patients with COVID-19 is associated with compositional changes in the airway microbiome and expansion of macrolide resistance genes. Taken together, our findings suggest that empiric macrolide use in patients with viral respiratory infections may contribute to antimicrobial resistance and public health risks, reinforcing the importance of stewardship efforts.

## Methods

### Study design, clinical cohort and ethics

IMPACC is a prospective longitudinal study that enroled 1,164 patients hospitalized for COVID-19 (refs. 23–25,38,39) from 20 hospitals across 15 academic biomedical centres within the United States between May 2020 and March 2021. Inclusion criteria included: (1) age ≥18 years; (2) confirmed SARS-CoV-2 positivity by reverse transcription PCR (RT–PCR) testing; and (3) confirmed understanding by participant and/ or surrogate of the data to be collected and the study procedures, and willingness to participate in the surveillance cohort. Exclusion criteria included: (1) underlying medical problems which, in the opinion of the investigator, may have been associated with mortality unrelated to COVID-19 within 48 h of hospitalization; (2) a decision by the patient or surrogate before hospitalization to limit care to comfort measures; or (3) medical problems or conditions such as pregnancy which might impact interpretation of the immunologic data obtained.

The Department of Health and Human Services Office for Human Research Protections (OHRP) and NIAID concurred that the IMPACC study qualified for public health surveillance exemption. The study protocol was reviewed by each site's institutional review board (IRB), with 12 sites conducting it as a public health surveillance study, and three sites integrating the IMPACC study into IRB-approved protocols (The University of Texas at Austin, IRB 2020-04-0117; University of California San Francisco, IRB 20-30497; Case Western Reserve

University, IRB STUDY20200573) with participants providing informed consent. Participants enroled at sites operating as a public health surveillance study were provided information sheets describing the study including the samples to be collected and plans for analysis and data de-identification. Participants who requested not to participate after review of the study plan and information were not enroled. Participants were not compensated while hospitalized but were subsequently compensated for outpatient visits and surveys. This study was registered at clinicaltrials.gov (NCT04378777) and followed the Strengthening the Reporting of Observational Studies in Epidemiology (STROBE) guidelines.

No participants were vaccinated for SARS-CoV-2 at the time of enrolment or during their hospitalization. To better categorize patients into different COVID-19 severity groups, they were classified into one of five trajectory groups using latent class mixed modelling of the degree of respiratory illness and external oxygen administration[24].

### Metatranscriptomic sequencing

Mid-turbinate nasal swabs were collected within 72 h of hospital admission and at subsequent visits with target dates of 4, 7, 14, 21 and 28 days post hospital admission. We elected to analyse nasal swabs because they were available on nearly all patients in the cohort, spanning the full range of disease severity. In contrast, lower airway tracheal aspirate specimens would only be obtainable from critically ill patients requiring mechanical ventilation, who represented <30% of the cohort. Bronchoscopy was rarely performed at most study site hospitals due to strict infection control precautions during the study period with respect to aerosol-generating procedures. Finally, many patients are unable to produce sputum without saline induction, another aerosol-generating procedure, and sputum collection was not routinely performed at study site hospitals.

The nasal swabs were stored in 1 ml of Zymo-DNA/RNA shield reagent (Zymo Research) before RNA was extracted twice in parallel from 250 µl of sample. The RNA was then purified with the KingFisher Flex sample purification system (Thermo Fisher) and the quick DNA-RNA MagBead kit (Zymo Research). We used a rigorous protocol and an ultra-clean laboratory space to prevent contamination when processing all samples. This involved wearing personal protective equipment including a gown, sterile gloves and a face shield when working with samples, using RNaseZap (Ambion) to wipe down pipets and bench surfaces to remove RNase contamination, and additionally spraying with 70% ethanol to disinfect and further clean. In addition, samples were thawed by extraction batch, and all tubes and plates were kept sealed or covered at all times until they were utilized.

Extracted RNA then underwent rRNA depletion, cDNA synthesis and library construction using the Illumina Total Stranded RNA Prep with Ribo-Zero Plus kit, following manufacturer instructions, and automated on a Perkin Elmer Sciclone NGSx Workstation. Control samples (Thermo Fisher, AM7832) were also included to assess for and minimize interbatch variability. Before sequencing, libraries were quantified using the Quant-it dsDNA High Sensitivity assay (Invitrogen), fragment size profiles were assessed using an Agilent fragment analyser, and any samples with >4% adapter dimers were removed. Libraries, normalized to 10 nM, underwent paired-end 100-base-pair sequencing on an Illumina NovaSeq 6000 instrument using S4 flow cells, targeting 50 million reads per sample.

### Microbiome and resistome profiling

Metatranscriptomic data were processed using the open-source CZ ID (https://czid.org/) Illumina mNGS pipeline (Nasal Swab data: v.7.1)[40,41]. Host reads were first removed by subtractive alignment against the GRCh38 human reference genome[42] using STAR[43]. Adapters were then removed using Trimmomatic[44], low-quality reads filtered using PriceSeq[45] and low complexity reads filtered using the Lempel–Ziv– Welch algorithm. Duplicate reads were identified and compressed

using czid-dedup[40]. A final scrub of any remaining host reads was carried out by realignment to GRCh38 using Bowtie2 (ref. 46). After performing these filtering steps, the remaining microbial data were subsampled to 2 million total reads. Taxonomic classification was then carried out by aligning reads to the NCBI nucleotide database using Minimap2 (refs. 47,48). In parallel, short reads were assembled into contiguous sequences using (SPAdes)[49], which then underwent alignment against accessions from the identified taxa to improve mapping specificity[40,49].

Resistome profiles were generated using the CZ ID antimicrobial resistance pipeline (v.0.2), which leverages the RGI tool and the CARD database[41,50]. Following microbiome and resistome profiling with CZ ID, background and batch correction were performed to remove contaminants and adjust for batch effects (see below). To limit the contribution of spurious hits to downstream analyses, additional filtering of microbial taxa and ARGs was performed on a per sample basis. Microbial taxa were excluded if they did not meet the following quality control criteria: (1) ≥10 alignments to the NCBI nucleotide database, (2) ≥1 alignment to the NCBI NR database and (3) alignment length of ≥50 bases. ARGs were excluded if they had <5% read coverage breadth and were either: (1) detected in ≤5% of samples with ARGs or (2) had an average sequencing depth across the gene normalized per million reads of ≤1 (depth per million, DPM) and had ≤10 alignments to the CARD database.

## Background and batch correction

Twenty-six negative control samples consisting of double-distilled water were processed and sequenced alongside clinical samples in the IMPACC cohort to enable the characterization and subtraction of background contamination. Reads mapping to taxa and ARGs in these control samples retained for downstream analyses after the above-described quality control measures are tabulated in Supplementary Tables 5 and 6, respectively. The sequencing data generated from these samples were analysed using the CZ ID metagenomic and antimicrobial resistance pipelines as described above. Background and batch correction was performed on the microbiome and resistome datasets separately. A negative binomial model was used to model the distribution of reads of microbial taxa/ARGs in the negative controls. Mean and dispersion parameters were then fitted to these data. Mean estimates were generated for each batch:taxon or batch:ARG pair in the negative controls, where batch corresponds to the phase of the IMPACC study (1, 2, 3A or 3B). The MASS package (v.7.3.58.1) in R was used to generate a single dispersion parameter across all taxa/ARGs. P values were adjusted for multiple testing using the Benjamini–Hochberg false discovery rate (FDR) algorithm. Microbial taxa/ARGs that were present at a significantly higher abundance in participant samples than in negative controls (FDR < 0.1) were retained for downstream analyses.

## Single timepoint analyses

Participants were assigned to one of three groups: those who received Azithromycin (Azithro) ± other antibiotics, those who took only non-Azithromycin antibiotics (Other-Abx), or participants who did not receive antibiotics (No-Abx). Participants with only partially captured antibiotic start and stop dates were excluded from these analyses. The two timepoints studied for these groups were the 1 ± 1 days of exposure and 5 ± 1 days of exposure, applicable to the Azithromycin and Other-Abx groups. The day 5 ± 1 timepoint was chosen for clinical relevance based on the typical azithromycin treatment course of 5 days. The day 1 ± 1 timepoint was selected to capture samples following brief antibiotic exposure. Given that a third intermediate timepoint at day 3 ± 1 would have included overlapping samples, we instead assessed these intermediate timepoints through generalized additive mixed model (GAMM) analysis. To match the No-Abx samples to the Azithromycin and Other-Abx samples at each timepoint, the distribution of days of hospitalization across those groups was examined. On the basis

of the findings, we selected samples for the No-Abx group as follows: In the 1 ± 1 days group, the samples had no antibiotic exposure and up to 40 days of hospitalization. In the 5 ± 1 days group, the samples had no antibiotic exposure, and between 3–40 days of hospitalization. Samples that qualified for both timepoints were split evenly between timepoints. The number of 'No-Abx' samples was also rarified (50%) to make the numbers in each group more comparable. In addition to this sample matching process for the No-Abx group, days from hospital admission was included as a covariate in statistical models to further control for slight differences. For the metagenomic analysis, if an azithromycin or other-ABX participant had no antibiotic exposure by that sampling timepoint, it was moved to the no-ABX group to better capture differences in brief antibiotic administration in the day 1 ± 1 timepoint. A linear mixed-effects model (using the lme4 package) was used to calculate differences between the groups at both timepoints while using age quintile, sex, severity TG, days from hospitalization, and receipt of steroids as fixed effects in the model, in combination with the participant's enrolment site included as a random effect. For the only-Azithromycin group, the number of days of the six most common co-administered antibiotics (vancomycin, ceftriaxone, cefepime, piperacillin-tazobactam, doxycycline and meropenem) was also included as a fixed effect. Resulting P values were adjusted using the Benjamini–Hochberg FDR algorithm via the p_adjust() function of the stats (v.4.2.3) package.

## GAMM analyses

Various metrics were modelled over days of azithromycin exposure using the nlme (v.3.1-162), lme4 (v.1.1-35.5), lmerTest (v.3.1-3), gamm4 (v.0.2-6) and ggeffects (v.1.7.2) packages in R. Samples were limited to the first 10 days of hospitalization and the first 5 days of antibiotic exposure due to few patients with longer antibiotic courses. The GAMMs included fixed effects of severity TG, sex, age quintile of the participant, and whether they ever received steroids, in addition to a smoothed term for days of azithromycin usage and days from hospitalization, with participant as a mixed effect. In addition, days of administration for the six most prevalent antibiotics in the cohort aside from azithromycin (vancomycin, ceftriaxone, cefepime, piperacillin-tazobactam, doxycycline and meropenem) were also added to the models as smooth terms.

R model formula:

~ s(Azithromycin, bs = 'cr', k = 4)+s(Ceftriaxone, bs = 'cr', k = 4)+ s(Vancomycin, bs = 'cr', k = 4)+s(Cefepime, bs = 'cr', k = 4)+s(Meropenem, bs = 'cr', k = 4)+s(Piperacillin_Tazobactam, bs = 'cr', k = 4)+s(Doxycycline, bs = 'cr', k = 4)+s(event_date, bs = 'cr')+ trajectory_group + sex + discretized_admit_age_quantile + ever_steroids

## Microbiome diversity metrics

Alpha diversity (Shannon diversity index) was calculated using the diversity() function in the vegan (2.6-6.1) R package. Beta diversity (Bray–Curtis dissimilarity) analysis was performed using the vegan functions vegdist(), betadisper(), permutest() and adonis2(). The beta diversity analysis was adjusted for age quintile, sex, days since hospitalization, severity TG, patient, and receipt of corticosteroids using the adonis2() function. For the resistome analysis, receipt of the six most common antibiotics aside from azithromycin (vancomycin, ceftriaxone, cefepime, piperacillin-tazobactam, doxycycline and meropenem) was also included in the model. Principal coordinate analysis (PCoA) of the resistome and bacterial microbiome was also performed using the cmdscale() function of the stats (v.4.2.3) package.

## Differential abundance analysis

Bacterial microbiome profile data were converted into a phyloseq object using the R packages ape (v.5.8.1), ade4 (v.1.7.22) and phyloseq (v.1.34.0). Differential abundance analysis was performed using the R package ANCOMBC (v.1.0.5) with an alpha level of 0.05 and prevalence filter (zero_cut argument) of 0.97. The analysis was adjusted for the

following covariates: age quintile, sex, days from hospitalization, severity TG, patient, receipt of corticosteroids, and receipt of the six most prevalent antibiotics aside from azithromycin (vancomycin, ceftriaxone, cefepime, piperacillin-tazobactam, doxycycline and meropenem). P values were adjusted for multiple testing using the Benjamini–Hochberg correction.

### Correlation analyses and pathogen-of-origin prediction

The 30 most abundant bacterial taxa by reads per million (RPM) were included in the correlation analysis. RPM was used as an abundance metric in lieu of raw reads to adjust for variations in sequencing depth between samples. The abundance metric used for ARGs was DPM, which adjusts for both variations in sequencing depth between samples and variations in gene length between ARGs. Spearman correlation was performed between bacterial taxa RPM and ARG DPM using the cor() function in the stats (v.4.2.3) package and the cor_pmat() function in the rstatix (v.0.7.2) package in R. Results were visualized using the corrplot (v.0.95) package.

For a subset of ARGs, the CARD RGI tool predicts pathogen of origin[31,50] by mapping ARG sequence *k*-mers to a database of 400+ primarily pathogenic taxa, providing information on both the taxon and the likely genomic location (for example, plasmid) of the ARG. Taxon assignments identified using this method were correlated with specific MLS ARGs utilizing the sum DPM among all samples with any exposure to azithromycin. These results were visualized using the pheatmap (v.1.0.13) package in R.

### Functional profiling and analysis

Microbiome data were functionally annotated using HUMAnN3 (ref. 27) and renormalized using the humann_renorm_table function. MaAsLin2 (ref. 28) was used to identify differentially abundant pathways in the normalized relative abundance data, with a prevalence cut-off of 1%, a minimum abundance cut-off of zero and log transformation.

### Peripheral blood mononuclear cell and nasal swab transcriptional profiling

Peripheral blood mononuclear cells (PBMCs) were isolated from 10 ml of whole blood collected by routine venipuncture which met minimal risk guidelines for hospitalized adults. RNA was extracted from $2.5 \times 10^5$ PBMCs lysed in Qiagen RLT buffer using the Quick-RNA MagBead kit (Zymo) with DNase digestion. RNA (10 ng) underwent library preparation using the SMART-Seq v4 Ultra Low Input RNA kit (Takara Bio) and the NexteraXT DNA Library Preparation kit (Illumina). Libraries underwent 100-base-pair, paired-end Illumina sequencing.

RNA-seq and alignment against the host transcriptome was performed following an established protocol[24,25] and the de-identified, quality-controlled raw gene count files and metadata were obtained from the IMPACC study. In this protocol, adapter trimming and quality filtering (minimum base quality score of Q30) were first carried out using Trimmomatic (v.0.36.5)[44]. STAR (v.2.4.2a)[43] was then used to align reads against the GRCh38 human reference genome[42] and gene counts were generated using HTSeq-count (v.0.4.1)[51]. Samples with a median coefficient of variation in gene coverage >0.8 or <1 million aligned counts were excluded from downstream analysis. We also filtered to retain only host protein-coding genes with at least 10 counts in at least 20% of the samples, together leaving 217 nasal and 238 PBMC samples available for analysis.

To identify genes that were associated with azithromycin exposure compared to No-Abx, we utilized the limma (v.3.58.1) package in R. We modelled gene expression as a function of azithromycin exposure, corrected for age, sex, trajectory group, days since hospitalization and exposure to corticosteroids. Then, we applied voom and duplicateCorrelation for two iterations to calculate the correlation between multiple samples from the same patients. Next, we applied the lmFit and eBayes functions to estimate the effects of azithromycin exposure on gene

expression and their P values. Finally, the P values were adjusted with Benjamini–Hochberg correction.

### Statistics

All code was written in R v.4.2.3 or R 4.0.3. Data processing was performed using the R packages dplyr (v.1.1.4) and tidyverse (v.2.0.0). Plots were generated using the R packages ggplot2 (v.3.5.1), ggpubr (v.0.6.0), scales (1.3.0) and ggbeeswarm (v.0.7.2). Package versions for each analysis are reported in the code repository.

### Reporting summary

Further information on research design is available in the Nature Portfolio Reporting Summary linked to this article.

## Data availability

Data used in this study are available at ImmPort Shared Data under the accession number SDY1760 and in the NLM's Database of Genotypes and Phenotypes (dbGaP) under the accession number phs002686.v2.p2. Source data for each figure are provided with this paper.

## Code availability

All code is deposited in Bitbucket at https://bitbucket.org/kleinstein/impacc-public-code/src/master/azithromycin_manuscript/ (ref. 52).

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

## Acknowledgements

We thank the participants of the study for their voluntary enrolment and contribution of samples for this work. See the supplement for details on the IMPACC Network. We acknowledge the assistance of the following individuals: S. Thomas, M. Cooney, S. Rao, S. Vignolo and E. Morrocchi (all from the CDCC); A. Naeim, M. Bernardo, S. Sanchez, S. Intluxay, C. Magyar, J. Brook, E. Ramires-Sanchez, M.

Llamas, C. Perdomo, C. E. Magyar and J. A. Fulcher (all from the David Geffen School of Medicine at UCLA); members of the UCLA Centre for Pathology Research Services and the Pathology Research Portal; M. C. Muenker, D. Duvilaire, M. Kuang, W. Ruff, K. Raddassi, D. Shepherd, H. Wang, O. Chaudhary, S. Salahuddin, J. Fournier and M. Rainone (all from the Yale School of Medicine). We thank the leadership of Boston Children's Hospital including W. Chung, G. Fleisher, N. Andrews and K. Churchwell for their support for the Precision Vaccines Program. The co-authorship of A.D.A. and P.M.B. of this report does not necessarily represent the official views of the National Institute of Allergy and Infectious Diseases, the National Institutes of Health (NIH) or any other agency of the US Government. Funding for this study was provided through the following US NIH grants: 5R01AI135803-03, 5U19AI118608-04, 5U19AI128910-04, 4U19AI090023-11, 4U19AI118610-06, R01AI145835-01A1S1, 5U19AI062629-17, 5U19AI057229-17, 5U19AI125357-05, 5U19AI128913-03, 3U19AI077439-13, 5U54AI142766-03, 5R01AI104870-07, 3U19AI089992-09, 3U19AI128913-03, 5T32DA018926-18 and K0826161611; NIAID, NIH grants 3U19AI1289130, U19AI128913-04S1, R01AI122220 and 1K23AI185326-01; NCATS (UM1TR004528); and the National Science Foundation (DMS2310836). Funding sources did not have a direct role in the design, analysis, or approval of this paper.

## Author contributions

C.R.L., A. Glascock, C. Maguire and V.T.C. conceived of the idea for the project. A. Glascock, C. Maguire and H.V.P. analysed the data. The IMPACC Network contributed to cohort design, participant enrolment, sample collection, data generation and/or data quality assurance. N.D.J. and M.A.A. generated the nasal metatranscriptomic data. C.R.L., A. Glascock, C. Maguire, H.V.P. and V.T.C. wrote the paper. A. Glascock, C. Maguire, H.V.P., V.T.C., J. Schaenman, O.L., S.H.K., E.F.R., J.D.-A., N. Rouphael and C.R.L. provided input on analyses and findings. A. Glascock, C. Maguire, H.V.P., E.C.L., J. Schaenman, C.S.C., E. Melamed, J.G., D.B.C., F. Krammer, L.R.B., R. Sekaly, G.A.M., E.K.H., C.B.C., L.N.G., B. Pulendran, A.F.-S., V.S., J.P.M., N.I.A.H., W.B.M., M.M.D., K.C.N., M.K., C. Bime, D.J.E., M.A.A., S.C.B., L.I.R.E., R.R.M., A.C.S., C.L.H., D.A.H., A.D.A., P.M.B., B. Peters, A.O., A.H., S.K.-S., F. Krammer, S. Bosinger, W.E., M.C.A., N.D.J., M.W., L. Guan, H.M., H.S., the IMPACC Network, J.D.-A., N. Rouphael, S.H.K., E.F.R., O.L., V.T.C. and C.R.L. reviewed and edited the paper.

## Competing interests

The Icahn School of Medicine at Mount Sinai has filed patent applications relating to SARS-CoV-2 serological assays, NDV-based SARS-CoV-2 vaccines, influenza virus vaccines and influenza virus therapeutics, which list F. Krammer as co-inventor and F. Krammer has received royalty payments from some of these patents. V.S. is listed on the SARS-CoV-2 serology assay patent. Mount Sinai has spun out a company, Castlevax, to develop SARS-CoV-2 vaccines. F. Krammer is co-founder and scientific advisory board member of Castlevax. F. Krammer has also consulted for Merck, GSK, Sanofi, Gritstone, Curevac, Seqirus and Pfizer, and is currently consulting for 3rd Rock Ventures and Avimex. The Krammer laboratory is also collaborating with Dynavax on influenza vaccine development. O.L. is a named inventor on patents held by Boston Children's Hospital relating to vaccine adjuvants and human in vitro platforms that model vaccine action. His laboratory has received research support from GlaxoSmithKline (GSK) and he is a co-founder of and advisor to ARMR Sciences (formerly Ovax, Inc.) that develops technologies to detect illicit substances and prevent overdose. He is also an advisor to GSK. C.B.C. serves as a consultant to bioMerieux and is funded by a grant from the Bill and Melinda Gates Foundation. J.A.O. is a consultant at Knocean Inc. J.L.-S. serves as a scientific advisor of Precion Inc. S.R.H., G.M. and K.W. are employees of Metabolon Inc. V.S.-M. is a current employee of MyOwnMed. N. Rouphael reports grants or contracts with Merck, Sanofi, Pfizer, Vaccine Company, Quidel, Lilly, and Immorna, and has participated on data safety monitoring

boards for Moderna, Sanofi, Seqirus, Pfizer, EMMES, ICON, BARDA, Imunon, CyanVac, and Micron. N. Rouphael has also received support for meetings/travel from Sanofi and Moderna, and honoraria from Virology Education. A.R. is a current employee of Immunai Inc. S.H.K. is a consultant related to ImmPort data repository for Peraton. D.A.H. has received research funding from Bristol–Myers Squibb, Novartis, Sanofi, and Genentech. He has been a consultant for Bayer Pharmaceuticals, Repertoire Inc., Bristol–Myers Squibb, Compass Therapeutics, EMD Serono, Genentech, Novartis Pharmaceuticals, and Sanofi Genzyme. A.I. is a consultant for 4BIO, Blue Willow Biologics, Revelar Biotherapeutics, RIGImmune, Xanadu Bio, and Paratus Sciences. M.K. receives research funds paid to her institution from NIH, ALA, Sanofi, and Astra-Zeneca for work in asthma; serves as a consultant for Astra-Zeneca, Sanofi, Chiesi, and GSK for severe asthma; and is a co-founder and CMO for RaeSedo, Inc., a company created to develop peptidomimetics for treatment of inflammatory lung disease. E. Melamed received research funding from Babson Diagnostics and honorarium from the Multiple Sclerosis Association of America, and has served on the advisory boards of Genentech, Horizon, Teva, and Viela Bio. C.C. receives research funding from NIH, FDA, DOD, Roche-Genentech and Quantum Leap Healthcare Collaborative, and also provides consulting services for Janssen, Vasomune, Gen1e Life Sciences, NGMBio, and Cellenkos. G.A.M. received research grants from Redhill, Cognivue, Pfizer, and Genentech, and served as a research consultant for Gilead, Merck, Viiv/GSK, and Jenssen. L.N.G. received research funding paid to her institution from Pfizer, Inc. E.F.R. serves as a consultant for Biogen and Regeneron for clinical trials advancing therapies in transplant alloimmunity. The other authors declare no competing interests.

## Additional information

**Extended data** is available for this paper at https://doi.org/10.1038/s41564-026-02285-8.

**Correspondence and requests for materials** should be addressed to Charles R. Langelier.

Abigail Glascock[1,42], Cole Maguire [2,42], Hoang Van Phan[3], Emily C. Lydon[3], Joanna Schaenman[4], Carolyn S. Calfee[3], Esther Melamed[2], John Greenland [3], David B. Corry [5], Farrah Kheradmand [5], Lindsey R. Baden [6], Rafick Sekaly[7], Grace A. McComsey [7], Elias K. Haddad [8], Charles B. Cairns [8], Linda N. Geng [9], Bali Pulendran [9], Ana Fernandez-Sesma [10], Viviana Simon [10,11,12,13,14], Jordan P. Metcalf[15], Nelson I. Agudelo Higuita [15], William B. Messer [16], Mark M. Davis [9], Kari C. Nadeau [9], Monica Kraft[17], Christian Bime[17], David J. Erle [3], Mark A. Atkinson [18], Scott C. Brakenridge[18], Lauren I. R. Ehrlich [2], Ruth R. Montgomery [19], Albert C. Shaw[19], Catherine L. Hough[16], David A. Hafler [19,20], Alison D. Augustine[21], Patrice M. Becker [21], Bjoern Peters [22], Al Ozonoff [20,23,24], Annmarie Hoch[25], Seunghee Kim-Schulze [26], Florian Krammer [10,11,14,27,28], Steven Bosinger [29], Walter Eckalbar[3], Matthew C. Altman [30], Michael Wilson [3], Leying Guan [31], Holden Maecker [9], Hanno Steen [23,24], IMPACC Network*, Joann Diray-Arce [24,25], Nadine Rouphael[29], Steven H. Kleinstein [31], Naresh Doni Jayavelu [30], Elaine F. Reed[4], Ofer Levy [20,23,24], Victoria T. Chu [3] & Charles R. Langelier [1,3] ✉

[1]Chan Zuckerberg Biohub San Francisco, San Francisco, CA, USA. [2]University of Texas at Austin, Austin, TX, USA. [3]University of California San Francisco, San Francisco, CA, USA. [4]University of California Los Angeles, Los Angeles, CA, USA. [5]Baylor College of Medicine and the Center for Translational Research on Inflammatory Diseases, Houston, TX, USA. [6]Brigham and Women's Hospital, Harvard Medical School, Boston, MA, USA. [7]Case Western Reserve University and University Hospitals of Cleveland, Cleveland, OH, USA. [8]Drexel University, Tower Health Hospital, Philadelphia, PA, USA. [9]Stanford University School of Medicine, Palo Alto, CA, USA. [10]Department of Microbiology, Icahn School of Medicine at Mount Sinai, New York, NY, USA. [11]Center for Vaccine Research and Pandemic Preparedness (C-VaRPP), Icahn School of Medicine at Mount Sinai, New York, NY, USA. [12]The Global Health and Emerging Pathogens Institute, Icahn School of Medicine at Mount Sinai, New York, NY, USA. [13]Division of Infectious Diseases, Department of Medicine, Icahn School of Medicine at Mount Sinai, New York, NY, USA. [14]Department of Pathology, Molecular and Cell-Based Medicine, Icahn School of Medicine at Mount Sinai, New York, NY, USA. [15]Oklahoma University Health Sciences Center, Oklahoma City, OK, USA. [16]Oregon Health and Science University, Portland, OR, USA. [17]University of Arizona, Tucson, AZ, USA. [18]University of Florida, Gainesville, FL, USA. [19]Yale School of Medicine, New Haven, CT, USA. [20]Broad Institute of MIT and Harvard, Cambridge, MA, USA. [21]National Institute of Allergy and Infectious Diseases, National Institute of Health, Bethesda, MD, USA. [22]La Jolla Institute for Immunology, La Jolla, CA, USA. [23]Precision Vaccines Program, Boston Children's Hospital, Harvard Medical School, Boston, MA, USA. [24]Boston Children's Hospital and Harvard Medical School, Boston, MA, USA. [25]Clinical and Data Coordinating Center (CDCC) Precision Vaccines Program, Boston Children's Hospital, Boston, MA, USA. [26]Icahn School of Medicine at Mount Sinai, New York, NY, USA. [27]Ignaz Semmelweis Institute, Interuniversity Institute for Infection Research, Medical University of Vienna, Vienna, Austria. [28]Ludwig Boltzmann Institute for Science Outreach and Pandemic Preparedness at the Medical University of Vienna, Vienna, Austria. [29]Emory School of Medicine, Atlanta, GA, USA. [30]Benaroya Research Institute, University of Washington, Seattle, WA, USA. [31]Yale School of Public Health, New Haven, CT, USA. [42]These authors contributed equally: Abigail Glascock, Cole Maguire. *A list of authors and their affiliations appears at the end of the paper. ✉e-mail: chaz.langelier@ucsf.edu

## IMPACC Network

Patrice M. Becker[21], Alison D. Augustine[21], Steven M. Holland[21], Lindsey B. Rosen[21], Serena Lee[21], Tatyana Vaysman[21], Al Ozonoff[25], Joann Diray-Arce[25], Jing Chen[25], Alvin Kho[25], Carly E. Milliren[25], Annmarie Hoch[25], Ana C. Chang[25], Kerry McEnaney[25], Brenda Barton[25], Claudia Lentucci[25], Maimouna D. Murphy[25], Mehmet Saluvan[25], Tanzia Shaheen[25], Shanshan Liu[25], Caitlin Syphurs[25], Marisa Albert[25], Arash Nemati Hayati[25], Robert Bryant[25], James Abraham[25], Sanya Thomas[25], Mitchell Cooney[25], Meagan Karoly[25], Matthew C. Altman[30], Naresh Doni Jayavelu[30], Scott Presnell[30], Bernard Kohr[30], Tomasz Jancsyk[30], Azlann Arnett[30], Bjoern Peters[22], James A. Overton[22,32], Randi Vita[22], Kerstin Westendorf[22], Ofer Levy[23], Hanno Steen[23], Patrick van Zalm[23], Benoit Fatou[23], Kinga K. Smolen[23], Arthur Viode[23], Simon van Haren[23], Meenakshi Jha[23], David Stevenson[23], Oludare Odumade[23], Lindsey R. Baden[6], Kevin Mendez[6], Jessica Lasky-Su[6], Alexandra Tong[6], Rebecca Rooks[6], Michael Desjardins[6], Amy C. Sherman[6], Stephen R. Walsh[6], Xhoi Mitre[6], Jessica Cauley[6], Xiofang Li[6], Bethany Evans[6], Christina Montesano[6], Jose Humberto Licona[6], Jonathan Krauss[6], Nicholas C. Issa[6], Jun Bai Park Chang[6], Natalie Izaguirre[6], Scott R. Hutton[33], Greg Michelotti[33], Kari Wong[33], Scott J. Tebbutt[34], Casey P. Shannon[34], Rafick-Pierre Sekaly[7], Slim Fourati[7], Grace A. McComsey[7], Paul Harris[7], Scott Sieg[7], Susan Pereira Ribeiro[7,29], Charles B. Cairns[8], Elias K. Haddad[8], Michele A. Kutzler[8], Mariana Bernui[8], Gina Cusimano[8], Jennifer Connors[8], Kyra Woloszczuk[8], David Joyner[8], Carolyn Edwards[8], Edward Lee[8], Edward Lin[8], Nataliya Melnyk[8], Debra L. Powell[8], James N. Kim[8], I. Michael Goonewardene[8], Brent Simmons[8], Cecilia M. Smith[8], Mark Martens[8], Brett Croen[8], Nicholas C. Semenza[8], Mathew R. Bell[8], Sara Furukawa[8], Renee McLin[8], George P. Tegos[8], Brandon Rogowski[8], Nathan Mege[8], Kristen Ulring[8], Pam Schearer[8], Judie Sheidy[8], Crystal Nagle[8], Vicki Seyfert-Margolis[35], Nadine Rouphael[29], Steven E. Bosinger[29], Arun K. Boddapati[29], Greg K. Tharp[29], Kathryn L. Pellegrini[29], Brandi Johnson[29], Bernadine Panganiban[29], Christopher Huerta[29], Evan J. Anderson[29], Hady Samaha[29], Jonathan E. Sevransky[29], Laurel Bristow[29], Elizabeth Beagle[29], David Cowan[29], Sydney Hamilton[29], Thomas Hodder[29], Amer Bechnak[29], Andrew Cheng[29], Aneesh Mehta[29], Caroline R. Ciric[29], Christine Spainhour[29], Erin Carter[29], Erin M. Scherer[29], Jacob Usher[29], Kieffer Hellmeister[29], Laila Hussaini[29], Lauren Hewitt[29], Nina Mcnair[29], Sonia Wimalasena[29], Ana Fernandez-Sesma[26], Viviana Simon[26], Florian Krammer[26], Harm Van Bakel[26], Seunghee Kim-Schulze[26], Ana Silvia Gonzalez Reiche[26], Jingjing Qi[26], Brian Lee[26], Juan Manuel Carreño[26], Gagandeep Singh[26], Ariel Raskin[26], Johnstone Tcheou[26], Zain Khalil[26], Adriana van de Guchte[26], Keith Farrugia[26], Zenab Khan[26], Geoffrey Kelly[26], Komal Srivastava[26], Lily Q. Eaker[26], Maria C. Bermúdez-González[26],

Lubbertus C. F. Mulder[26], Katherine F. Beach[26], Miti Saksena[26], Deena Altman[26], Erna Kojic[26], Levy A. Sominsky[26], Arman Azad[26], Dominika Bielak[26], Hisaaki Kawabata[26], Temima Yellin[26], Miriam Fried[26], Leeba Sullivan[26], Sara Morris[26], Giulio Kleiner[26], Daniel Stadlbauer[26], Jayeeta Dutta[26], Hui Xie[26], Manishkumar Patel[26], Kai Nie[26], Adeeb Rahman[36], William B. Messer[37], Catherine L. Hough[37], Sarah A. R. Siegel[37], Peter E. Sullivan[37], Zhengchun Lu[37], Amanda E. Brunton[37], Matthew Strand[37], Zoe L. Lyski[37], Felicity J. Coulter[37], Courtney Micheleti[37], Holden Maecker[9], Bali Pulendran[9], Kari C. Nadeau[9], Yael Rosenberg-Hasson[9], Michael Leipold[9], Natalia Sigal[9], Angela Rogers[9], Andrea Fernandes[9], Monali Manohar[9], Evan Do[9], Iris Chang[9], Alexandra S. Lee[9], Catherine Blish[9], Henna Naz Din[9], Jonasel Roque[9], Linda Geng[9], Maja Artandi[9], Mark M. Davis[9], Neera Ahuja[9], Samuel S. Yang[9], Sharon Chinthrajah[9], Thomas Hagan[9], Elaine F. Reed[38], Joanna Schaenman[38], Ramin Salehi-Rad[38], Adreanne M. Rivera[38], Harry C. Pickering[38], Subha Sen[38], David Elashoff[38], Dawn C. Ward[38], Jenny Brook[38], Estefania Ramires Sanchez[38], Megan Llamas[38], Claudia Perdomo[38], Clara E. Magyar[38], Jennifer Fulcher[38], David J. Erle[3], Carolyn S. Calfee[3], Carolyn M. Hendrickson[3], Kirsten N. Kangelaris[3], Viet Nguyen[3], Deanna Lee[3], Suzanna Chak[3], Rajani Ghale[3], Ana Gonzalez[3], Alejandra Jauregui[3], Carolyn Leroux[3], Luz Torres Altamirano[3], Ahmad Sadeed Rashid[3], Andrew Willmore[3], Prescott G. Woodruff[3], Matthew F. Krummel[3], Sidney Carrillo[3], Alyssa Ward[3], Charles R. Langelier[3], Ravi Patel[3], Michael Wilson[3], Ravi Dandekar[3], Bonny Alvarenga[3], Jayant Rajan[3], Walter Eckalbar[3], Andrew W. Schroeder[3], Gabriela K. Fragiadakis[3], Alexandra Tsitsiklis[3], Eran Mick[3], Yanedth Sanchez Guerrero[3], Christina Love[3], Lenka Maliskova[3], Michael Adkisson[3], Aleksandra Leligdowicz[3], Alexander Beagle[3], Arjun Rao[3], Austin Sigman[3], Bushra Samad[3], Cindy Curiel[3], Cole Shaw[3], Gayelan Tietje-Ulrich[3], Jeff Milush[3], Jonathan Singer[3], Joshua J. Vasquez[3], Kevin Tang[3], Legna Betancourt[3], Lekshmi Santhosh[3], Logan Pierce[3], Maria Tecero Paz[3], Michael Matthay[3], Neeta Thakur[3], Nicklaus Rodriguez[3], Nicole Sutter[3], Norman Jones[3], Pratik Sinha[3], Priya Prasad[3], Raphael Lota[3], Sadeed Rashid[3], Saurabh Asthana[3], Sharvari Bhide[3], Tasha Lea[3], Yumiko Abe-Jones[3], David A. Hafler[19], Ruth R. Montgomery[19], Albert C. Shaw[19], Steven H. Kleinstein[19], Jeremy P. Gygi[19], Shrikant Pawar[19], Anna Konstorum[19], Ernie Chen[19], Chris Cotsapas[19], Xiaomei Wang[19], Leqi Xu[19], Charles Dela Cruz[19], Akiko Iwasaki[19], Subhasis Mohanty[19], Allison Nelson[19], Yujiao Zhao[19], Shelli Farhadian[19], Hiromitsu Asashima[19], Omkar Chaudhary[19], Andreas Coppi[19], John Fournier[19], M. Catherine Muenker[19], Allison Nelson[19], Khadir Raddassi[19], Michael Rainone[19], William Ruff[19], Syim Salahuddin[19], Wade L. Shulz[19], Pavithra Vijayakumar[19], Haowei Wang[19], Esio Wunder Jr.[19], H. Patrick Young[19], Albert I. Ko[19], Denise Esserman[31], Leying Guan[31], Anderson Brito[31], Jessica Rothman[31], Nathan D. Grubaugh[31], David B. Corry[5], Farrah Kheradmand[5], Li-Zhen Song[5], Ebony Nelson[5], Jordan P. Metcalf[15], Nelson I. Agudelo Higuita[15], Lauren A. Sinko[15], J. Leland Booth[15], Douglas A. Drevets[15], Brent R. Brown[15], Monica Kraft[17], Christian Bime[17], Jarrod Mosier[17], Heidi Erickson[17], Ron Schunk[17], Hiroki Kimura[17], Michelle Conway[17], Dave Francisco[17], Allyson Molzahn[17], Connie Cathleen Wilson[17], Trina Hughes[17], Bianca Sierra[17], Mark A. Atkinson[18], Scott C. Brakenridge[18], Ricardo F. Ungaro[18], Brittany Roth Manning[18], Jordan Oberhaus[39], Faheem W. Guirgis[39], Brittney Borresen[40], Matthew L. Anderson[40], Lauren I. R. Ehrlich[41], Esther Melamed[41], Cole Maguire[41], Dennis Wylie[41], Justin F. Rousseau[41], Kerin C. Hurley[41], Janelle N. Geltman[41], Nadia Siles[41], Jacob E. Rogers[41] & Pablo Guaman Tipan[41]

[32]Knocean Inc., Toronto, Ontario, Canada. [33]Metabolon Inc., Morrisville, NC, USA. [34]Prevention of Organ Failure (PROOF) Centre of Excellence, University of British Columbia, Vancouver, British Columbia, Canada. [35]MyOwnMed Inc., Bethesda, MD, USA. [36]Immunai Inc., New York, NY, USA. [37]Oregon Health Sciences University, Portland, OR, USA. [38]David Geffen School of Medicine at the University of California Los Angeles, Los Angeles, CA, USA. [39]Lyle Moldawer University of Florida, Jacksonville, FL, USA. [40]University of South Florida, Tampa, FL, USA. [41]The University of Texas at Austin, Austin, TX, USA.

## Extended Data Table 1 | Cohort Demographics

| | | Overall (N=1164) | No Antibiotics (n=474) | Azithromycin (n=366) | Other Antibiotics (n=324) | Overall p-value | No vs Azithro p-value | Other vs Azithro p-value |
|---|---|---|---|---|---|---|---|---|
| **Age at enrollment (years), median (IQR)** | (n=1164) | 59.0 (20.0) | 57.0 (21.0) | 59.0 (19.0) | 62.0 (19.0) | <.001 | 0.025 | 0.000 |
| **Sex at birth** | Male | 711 (61%) | 289 (61%) | 228 (62%) | 194 (60%) | 0.808 | 0.696 | 0.756 |
| | Female | 453 (39%) | 185 (39%) | 138 (38%) | 130 (40%) | | | |
| **Race** | White | 562 (48%) | 205 (43%) | 189 (52%) | 168 (52%) | 0.117 | 0.289 | 0.037 |
| | Black | 259 (22%) | 125 (26%) | 75 (20%) | 59 (18%) | | | |
| | Other | 210 (18%) | 93 (20%) | 62 (17%) | 55 (17%) | | | |
| | Asian | 51 (4%) | 18 (4%) | 19 (5%) | 14 (4%) | | | |
| | Multiple | 15 (1%) | 4 (1%) | 3 (1%) | 8 (2%) | | | |
| | American Indian/Alaska Native | 14 (1%) | 7 (1%) | 4 (1%) | 3 (1%) | | | |
| | Native Hawaiian/Pacific Islander | 11 (1%) | 6 (1%) | 3 (1%) | 2 (1%) | | | |
| | Unknown | 42 (4%) | 16 (3%) | 11 (3%) | 15 (5%) | | | |
| **Hispanic ethnicity** | Non-Hispanic | 753 (65%) | 320 (68%) | 218 (60%) | 215 (66%) | 0.020 | 0.008 | 0.917 |
| | Hispanic | 364 (31%) | 133 (28%) | 138 (38%) | 93 (29%) | | | |
| | Unknown | 47 (4%) | 21 (4%) | 10 (3%) | 16 (5%) | | | |
| **Comorbidities** | None | 68 (6%) | 31 (7%) | 18 (5%) | 19 (6%) | 0.039 | 0.173 | 0.114 |
| | Hypertension | 677 (58%) | 266 (56%) | 209 (57%) | 202 (62%) | 0.191 | 0.775 | 0.079 |
| | Diabetes | 427 (37%) | 161 (34%) | 138 (38%) | 128 (40%) | 0.249 | 0.262 | 0.110 |
| | Chronic lung disease | 235 (20%) | 86 (18%) | 67 (18%) | 82 (25%) | 0.026 | 0.952 | 0.015 |
| | Asthma | 174 (15%) | 75 (16%) | 48 (13%) | 51 (16%) | 0.493 | 0.271 | 0.975 |
| | Chronic cardiac disease | 315 (27%) | 140 (30%) | 77 (21%) | 98 (30%) | 0.007 | 0.005 | 0.829 |
| | Chronic kidney disease | 178 (15%) | 59 (12%) | 58 (16%) | 61 (19%) | 0.046 | 0.158 | 0.013 |
| | Chronic liver disease | 58 (5%) | 26 (5%) | 14 (4%) | 18 (6%) | 0.469 | 0.263 | 0.966 |
| | Chronic neurological disorder | 143 (12%) | 60 (13%) | 29 (8%) | 54 (17%) | 0.002 | 0.027 | 0.112 |
| | Organ Transplantation | 69 (6%) | 18 (4%) | 28 (8%) | 23 (7%) | 0.037 | 0.015 | 0.038 |
| | HIV/AIDS | 21 (2%) | 9 (2%) | 3 (1%) | 9 (3%) | 0.153 | 0.191 | 0.411 |
| | Malignancy | 119 (10%) | 44 (9%) | 25 (7%) | 50 (15%) | <.001 | 0.199 | 0.008 |
| **BMI Category** | Underweight | 15 (1%) | 7 (1%) | 4 (1%) | 4 (1%) | <.001 | 0.209 | 0.001 |
| | Normal weight | 166 (14%) | 55 (12%) | 40 (11%) | 71 (22%) | | | |
| | Overweight (25.1-29.9) | 300 (26%) | 115 (24%) | 100 (27%) | 85 (26%) | | | |
| | Class 1-2 Obesity (30-39.9) | 473 (41%) | 194 (41%) | 163 (45%) | 116 (36%) | | | |
| | Class 3 Obesity (40+) | 167 (14%) | 77 (16%) | 50 (14%) | 40 (12%) | | | |
| | Missing | 43 (4%) | 26 (5%) | 9 (2%) | 8 (2%) | | | |
| **Level of respiratory support** | Mechanically ventilated, or ECMO (OS=6) | 139 (12%) | 12 (3%) | 64 (17%) | 63 (19%) | <.001 | 0.000 | 0.000 |
| | Non-invasive ventilation, or high flow nasal O2 (OS=5) | 212 (18%) | 62 (13%) | 89 (24%) | 61 (19%) | | | |
| | Supplemental oxygen (not high flow) (OS=4) | 546 (47%) | 269 (57%) | 159 (43%) | 118 (36%) | | | |
| | None (OS=3) | 266 (23%) | 131 (28%) | 54 (15%) | 81 (25%) | | | |
| | Missing | 1 (0%) | 0 (0%) | 0 (0%) | 1 (0%) | | | |
| **Medications** | Steroids | 793 (68%) | 273 (58%) | 291 (80%) | 229 (71%) | <.001 | 0.000 | 0.000 |
| **Length of stay (days)** | (n=909) | 6.0 (8.0) | 5.0 (5.0) | 7.0 (9.0) | 8.0 (13.0) | <.001 | 0.000 | 0.000 |
| **Acute trajectory group** | 1 | 232 (20%) | 126 (27%) | 49 (13%) | 57 (18%) | <.001 | 0.000 | 0.000 |
| | 2 | 265 (23%) | 128 (27%) | 82 (22%) | 55 (17%) | | | |
| | 3 | 337 (29%) | 166 (35%) | 99 (27%) | 72 (22%) | | | |
| | 4 | 222 (19%) | 42 (9%) | 97 (27%) | 83 (26%) | | | |
| | 5 | 108 (9%) | 12 (3%) | 39 (11%) | 57 (18%) | | | |

ABX = antibiotics. CRP = C-reactive protein.

**Extended Data Table 2 | Antibiotic use across COVID-19 severity trajectory groups (TGs)**

| Antibiotic | Total (n=1154) | TG1 (n=232) | TG2 (n=265) | TG3 (n=337) | TG4 (n=222) | TG5 (n=108) |
|---|---|---|---|---|---|---|
| Amikacin | 0.09% (1) | 0% (0) | 0% (0) | 0.3% (1) | 0% (0) | 0% (0) |
| Ampicillin-Sulbactam | 0.86% (10) | 0% (0) | 0.75% (2) | 0.3% (1) | 2.7% (6) | 0.93% (1) |
| Ampicillin | 2.06% (24) | 2.16% (5) | 0.38% (1) | 0.89% (3) | 5.86% (13) | 1.85% (2) |
| Azithromycin | 29.81% (347) | 20.69% (48) | 29.06% (77) | 28.19% (95) | 40.54% (90) | 34.26% (37) |
| Aztreonam | 0.43% (5) | 0.43% (1) | 0.38% (1) | 0% (0) | 0.9% (2) | 0.93% (1) |
| Cefazolin | 2.23% (26) | 1.29% (3) | 0.75% (2) | 1.19% (4) | 4.95% (11) | 5.56% (6) |
| Cefepime | 12.71% (148) | 3.02% (7) | 5.28% (14) | 4.45% (15) | 31.08% (69) | 39.81% (43) |
| Cefpodoxime | 0.43% (5) | 1.29% (3) | 0% (0) | 0.59% (2) | 0% (0) | 0% (0) |
| Ceftazidime | 1.72% (20) | 0% (0) | 0.38% (1) | 0.3% (1) | 4.5% (10) | 7.41% (8) |
| Ceftriaxone | 36.77% (428) | 30.6% (71) | 35.09% (93) | 31.16% (105) | 49.55% (110) | 45.37% (49) |
| Cefuroxime | 0.26% (3) | 0% (0) | 0% (0) | 0.89% (3) | 0% (0) | 0% (0) |
| Ciprofloxacin | 1.03% (12) | 0.43% (1) | 0.38% (1) | 1.19% (4) | 2.25% (5) | 0.93% (1) |
| Clindamycin | 0.17% (2) | 0% (0) | 0% (0) | 0.59% (2) | 0% (0) | 0% (0) |
| Daptomycin | 0.26% (3) | 0% (0) | 0% (0) | 0% (0) | 0.9% (2) | 0.93% (1) |
| Doxycycline | 7.13% (83) | 6.9% (16) | 9.06% (24) | 5.34% (18) | 8.11% (18) | 6.48% (7) |
| Ertapenem | 1.55% (18) | 0% (0) | 0.75% (2) | 0.89% (3) | 4.5% (10) | 2.78% (3) |
| Erythromycin | 0.34% (4) | 0.43% (1) | 0% (0) | 0% (0) | 0.9% (2) | 0.93% (1) |
| Gentamicin | 0.09% (1) | 0% (0) | 0% (0) | 0% (0) | 0.45% (1) | 0% (0) |
| Levofloxacin | 2.32% (27) | 1.29% (3) | 0.75% (2) | 2.37% (8) | 4.95% (11) | 2.78% (3) |
| Linezolid | 0.77% (9) | 0.43% (1) | 0% (0) | 0% (0) | 2.25% (5) | 2.78% (3) |
| Meropenem | 3.87% (45) | 0% (0) | 0.75% (2) | 1.19% (4) | 12.16% (27) | 11.11% (12) |
| Metronidazole | 3.09% (36) | 0.86% (2) | 1.13% (3) | 1.19% (4) | 7.66% (17) | 9.26% (10) |
| Piperacillin-Tazobactam | 9.02% (105) | 3.02% (7) | 4.91% (13) | 3.86% (13) | 20.72% (46) | 24.07% (26) |
| Rifampin | 0.09% (1) | 0% (0) | 0% (0) | 0% (0) | 0.45% (1) | 0% (0) |
| Rifaximin | 0.43% (5) | 0.43% (1) | 0% (0) | 0% (0) | 1.35% (3) | 0.93% (1) |
| Sulfamethoxazole-Trimethoprim | 1.55% (18) | 0.43% (1) | 0.75% (2) | 1.48% (5) | 3.6% (8) | 1.85% (2) |
| Tobramycin | 0.34% (4) | 0% (0) | 0% (0) | 0% (0) | 1.35% (3) | 0.93% (1) |
| Vancomycin | 21.22% (247) | 4.74% (11) | 11.7% (31) | 10.39% (35) | 45.5% (101) | 63.89% (69) |
| Other | 3.09% (36) | 1.72% (4) | 2.64% (7) | 1.78% (6) | 5.86% (13) | 5.56% (6) |

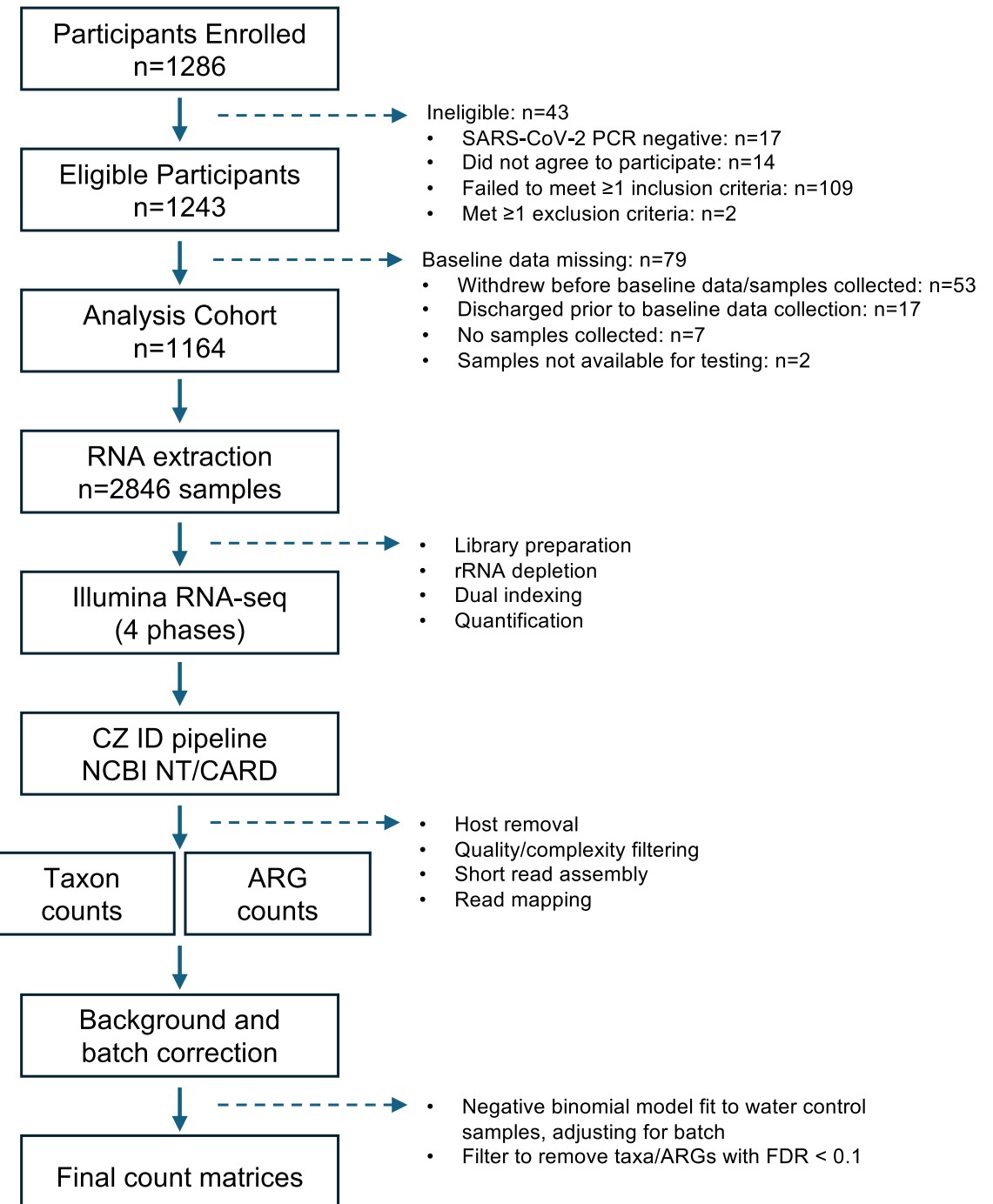

**Extended Data Fig. 1 | Study design and analysis flow diagram.** The IMPACC (Immunophenotyping Assessment in a COVID-19 Cohort) study enrolled 1286 participants hospitalized for COVID-19 across the United States. Biological samples from 1164 participants were ultimately available for analysis. RNA extraction and sequencing library preparation were carried out followed by Illumina RNA sequencing. Taxon and antimicrobial resistance gene (ARG) counts were generated by alignment against the NCBI NT and CARD databases, respectively, using the CZ ID bioinformatics pipeline. Correction and mitigation of background environmental contaminants was then carried out using a statistical model fit to data from water control samples, ultimately yielding final microbial taxon and ARG count matrices for downstream analyses.

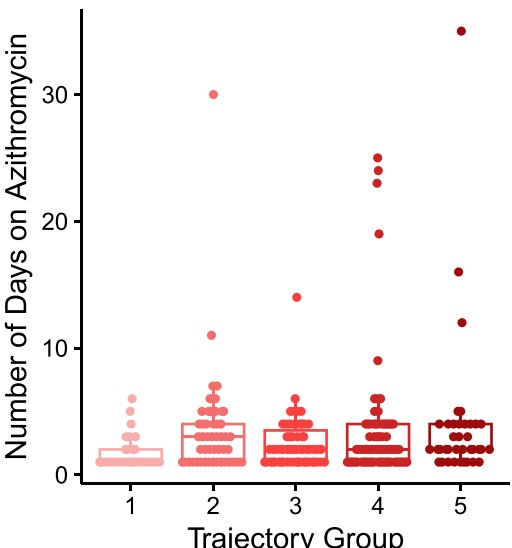

**Extended Data Fig. 2 | Days of azithromycin exposure for each participant by COVID-19 severity trajectory group.** Participants with zero days of azithromycin were omitted as points to visualize number of days when azithromycin was administered. Boxplot limits correspond to the interquartile range (IQR) and the center line the median. The lower whisker extends to the smallest value within 1.5 * IQR below Q1, and the upper whisker extends to the largest value within 1.5 * IQR above Q3. Sample size for participants in each trajectory group with at least one day of azithromycin administration: TG1 n = 27, TG2 n = 49, TG3 n = 59, TG4 n = 74, and TG5 n = 37.

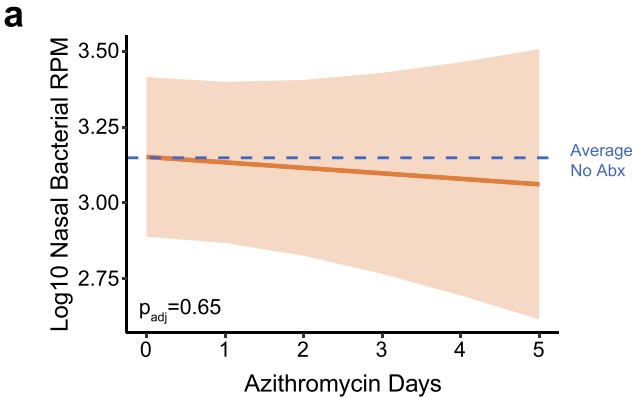

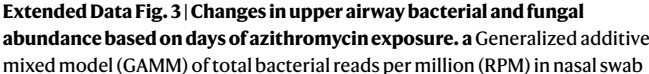

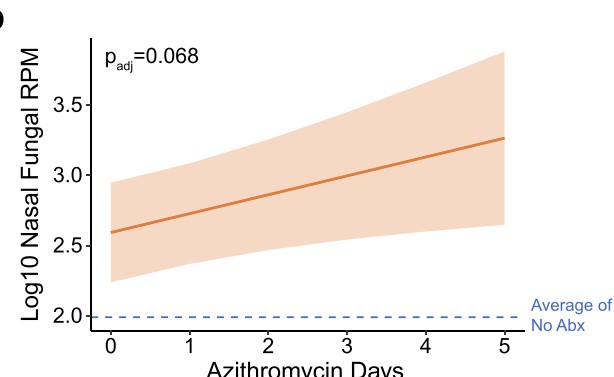

**Extended Data Fig. 3 | Changes in upper airway bacterial and fungal abundance based on days of azithromycin exposure. a** Generalized additive mixed model (GAMM) of total bacterial reads per million (RPM) in nasal swab samples by days of azithromycin usage. **b** Total fungal RPM in nasal swab samples by days of azithromycin usage. Shaded region denotes 95% confidence interval. P-values adjusted with Benjamini-Hochberg correction.

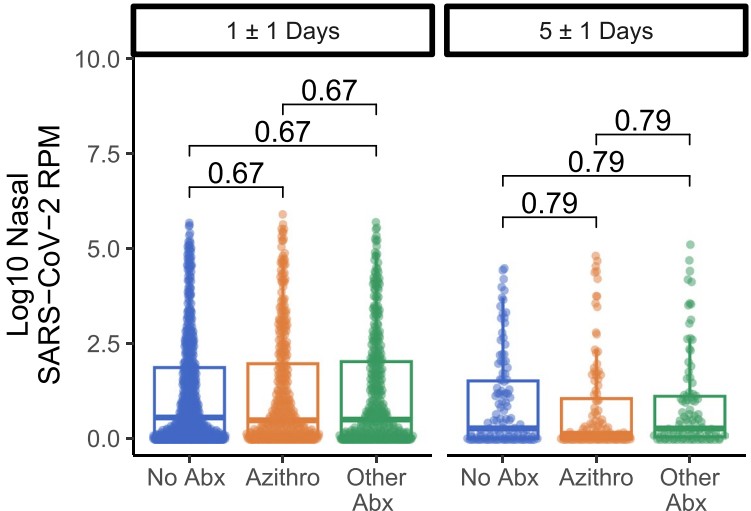

**Extended Data Fig. 4 | SARS-CoV-2 abundance (reads per million, RPM) does not differ based on azithromycin exposure.** Boxplot limits correspond to the interquartile range (IQR) and the center line the median. The lower whisker extends to the smallest value within 1.5 * IQR below Q1, and the upper whisker extends to the largest value within 1.5 * IQR above Q3. Significance calculated with a pairwise two-sided Wilcoxon rank-sum test with Benjamini-Hochberg corrections. Sample sizes for groups are: no ABX 1 ± 1 day n = 666, no ABX 5 ± 1 day n = 120, azithromycin 1 ± 1 day n = 394, azithromycin 5 ± 1 day n = 116, other ABX 1 ± 1 day n = 396, and other ABX 5 ± 1 day n = 99.

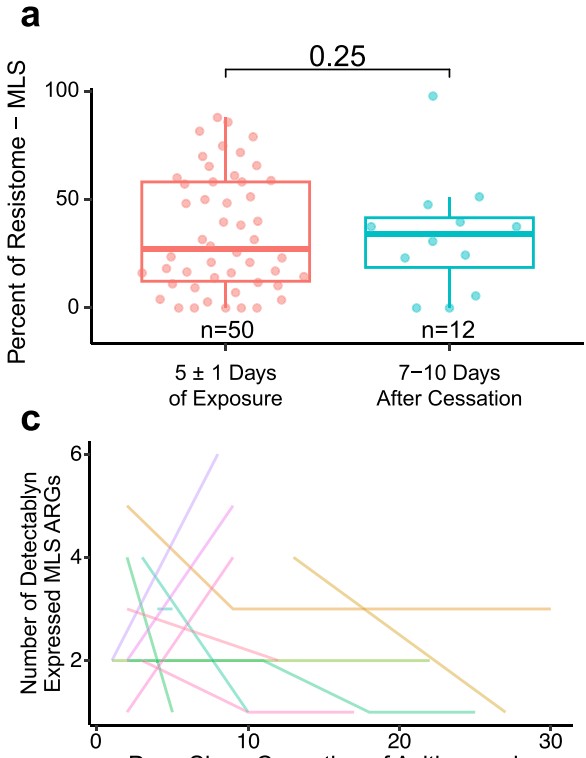

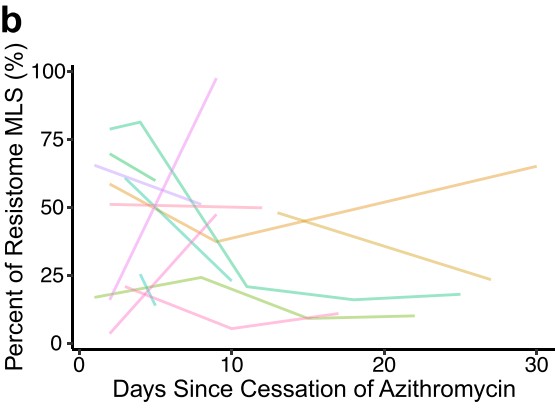

**Extended Data Fig. 5 | MLS ARG class metrics demonstrate no significant decrease after cessation of azithromycin. a**. MLS class percent of resistome before and after azithromycin cessation. P values are based on a linear mixed-effects model. Boxplots display the median, first and third quartile, and range. Data points are randomly jittered along the x-axis for visual clarity. **b**. Spaghetti plot of MLS percent of resistome over time following azithromycin cessation (n = 12). **c**. Spaghetti plot of MLS ARG richness over time following azithromycin cessation (n = 12). Colors represent different participants. Repeat measures from the same participant are connected. Boxplot limits correspond to the interquartile range (IQR) and the center line the median. The lower whisker extends to the smallest value within 1.5 * IQR below Q1, and the upper whisker extends to the largest value within 1.5 * IQR above Q3.

**a**

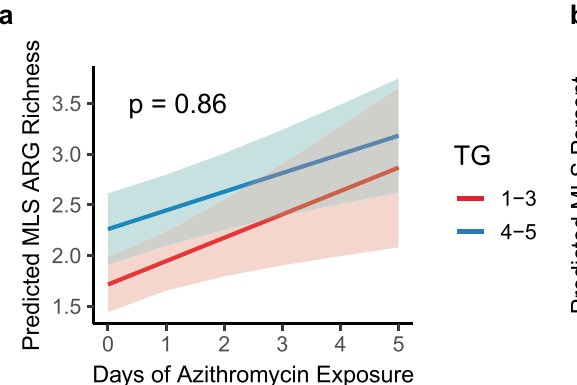

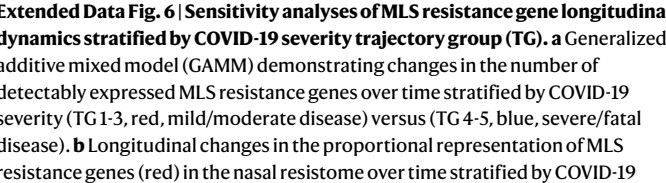

**b**

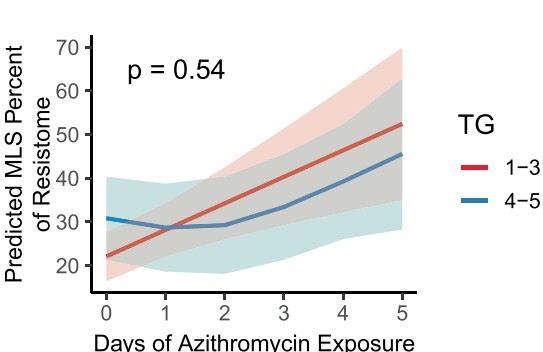

**Extended Data Fig. 6 | Sensitivity analyses of MLS resistance gene longitudinal dynamics stratified by COVID-19 severity trajectory group (TG). a** Generalized additive mixed model (GAMM) demonstrating changes in the number of detectably expressed MLS resistance genes over time stratified by COVID-19 severity (TG 1-3, red, mild/moderate disease) versus (TG 4-5, blue, severe/fatal disease). **b** Longitudinal changes in the proportional representation of MLS resistance genes (red) in the nasal resistome over time stratified by COVID-19 severity (TG 1-3, red, mild/moderate disease, n = 434) versus (TG 4-5, blue, severe/ fatal disease, n = 334). The smoothed curves represent the estimated non-linear relationship between variables. The center line represents the predicted mean value, and the shaded area denotes the 95% confidence interval. Global p-values were generated from a likelihood-ratio test comparing a non-stratified GAMM to a full GAMM with group-specific trajectories.

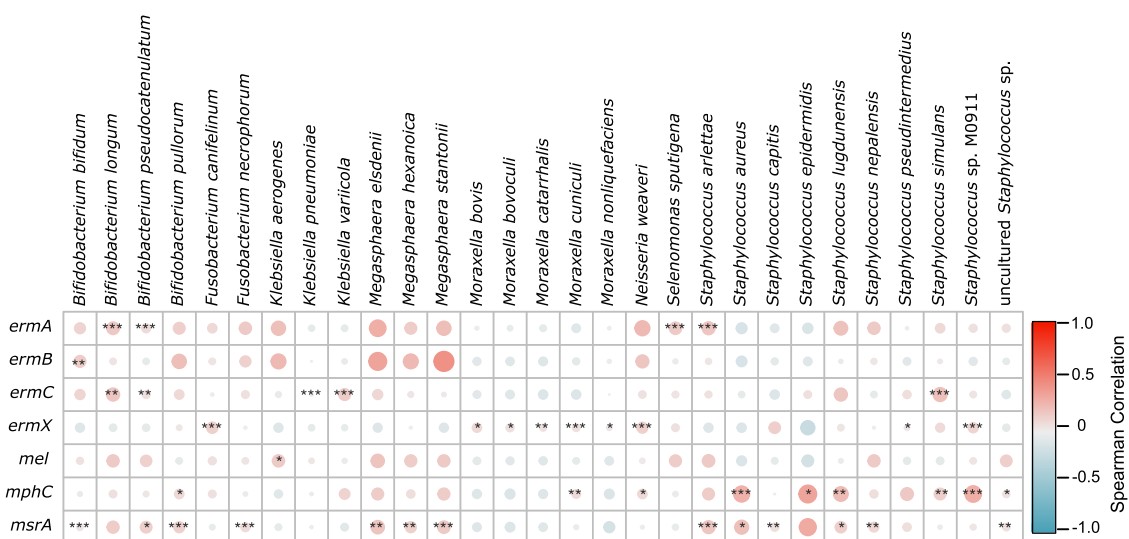

**Extended Data Fig. 7 | Species-level correlations within the airway resistome and microbiome.** Species-level Spearman correlation analysis among differentially abundant genera from Fig. 2f. Only taxa with significant correlations are displayed (n = 205). Color bar reflects two-sided Spearman correlation coefficient. ***$p_{adj}$ < 0.001; **$p_{adj}$ < 0.01; *$p_{adj}$ < 0.05.

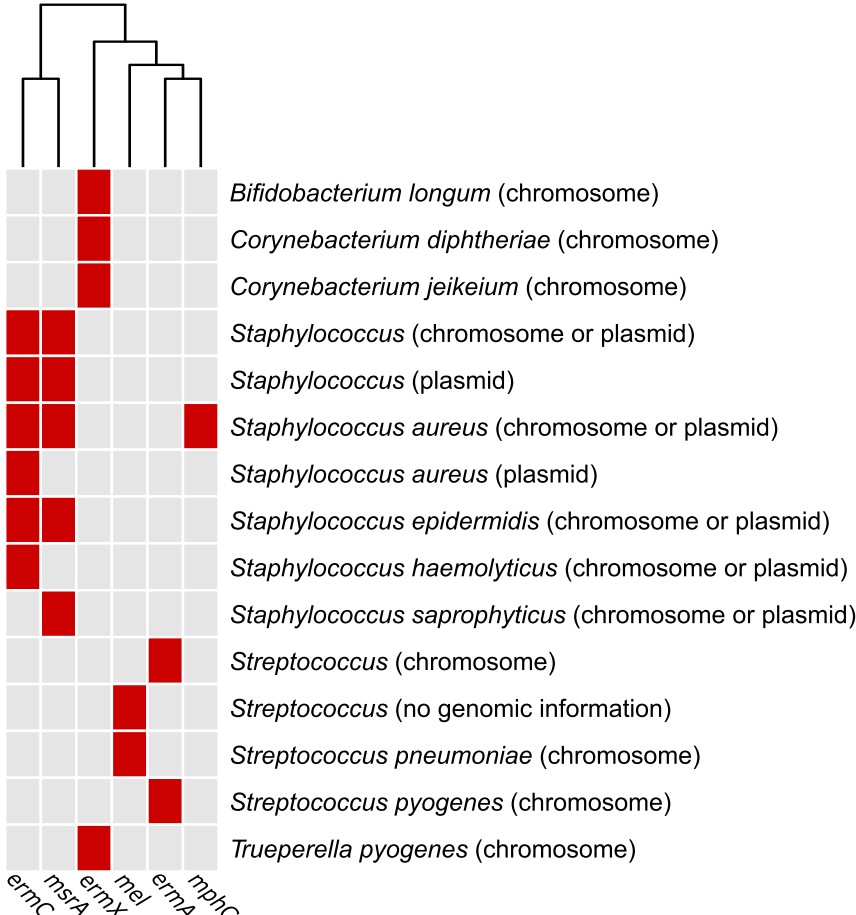

**Extended Data Fig. 8 | Species-level taxonomic assignment of MLS resistance genes based on CARD pathogen-of-origin assignments.** Heatmap of CARD pathogen-of-origin assignments and MLS ARGs (n = 205). Red cells denote the assignment of a given ARG to a given taxon by CARD. Genomic location information in parentheses.

# Reporting Summary

## Statistics

For all statistical analyses, confirm that the following items are present in the figure legend, table legend, main text, or Methods section.

| n/a | Confirmed | |
|---|---|---|
| ☐ | ☒ | The exact sample size (*n*) for each experimental group/condition, given as a discrete number and unit of measurement |
| ☐ | ☒ | A statement on whether measurements were taken from distinct samples or whether the same sample was measured repeatedly |
| ☐ | ☒ | The statistical test(s) used AND whether they are one- or two-sided *Only common tests should be described solely by name; describe more complex techniques in the Methods section.* |
| ☐ | ☒ | A description of all covariates tested |
| ☐ | ☒ | A description of any assumptions or corrections, such as tests of normality and adjustment for multiple comparisons |
| ☐ | ☒ | A full description of the statistical parameters including central tendency (e.g. means) or other basic estimates (e.g. regression coefficient) AND variation (e.g. standard deviation) or associated estimates of uncertainty (e.g. confidence intervals) |
| ☐ | ☒ | For null hypothesis testing, the test statistic (e.g. *F*, *t*, *r*) with confidence intervals, effect sizes, degrees of freedom and *P* value noted *Give P values as exact values whenever suitable.* |
| ☒ | ☐ | For Bayesian analysis, information on the choice of priors and Markov chain Monte Carlo settings |
| ☒ | ☐ | For hierarchical and complex designs, identification of the appropriate level for tests and full reporting of outcomes |
| ☐ | ☒ | Estimates of effect sizes (e.g. Cohen's *d*, Pearson's *r*), indicating how they were calculated |

*Our web collection on statistics for biologists contains articles on many of the points above.*

## Software and code

Policy information about availability of computer code

| | |
|---|---|
| Data collection | Metatranscriptomic data was processed using the open-source CZ ID pipeline (https://czid.org/). Microbiome profiling was performed within CZ ID using the Illumina mNGS pipeline (v7.1). Resistome profiles were generated using the CZ ID AMR pipeline (v0.2), which leverages the Resistance Gene Identifier (RGI) tool and the Comprehensive Antibiotic Resistance Database (CARD). |
| Data analysis | All code was written in R v4.2.3 or R 4.0.3. Data processing was performed using R packages dplyr (v1.1.4) and tidyverse (v2.0.0). Plots were generated using R packages ggplot2 (v3.5.1), ggpubr (v0.6.0), scales (1.3.0) and ggbeeswarm (v0.7.2). Package versions for each analysis are reported in the code repository. |

For manuscripts utilizing custom algorithms or software that are central to the research but not yet described in published literature, software must be made available to editors and reviewers. We strongly encourage code deposition in a community repository (e.g. GitHub). See the Nature Portfolio guidelines for submitting code & software for further information.

# Data

Policy information about availability of data

All manuscripts must include a data availability statement. This statement should provide the following information, where applicable:

- Accession codes, unique identifiers, or web links for publicly available datasets
- A description of any restrictions on data availability
- For clinical datasets or third party data, please ensure that the statement adheres to our policy

Data used in this study is available at ImmPort Shared Data under the accession number SDY1760 and in the NLM's Database of Genotypes and Phenotypes (dbGaP) under the accession number phs002686.v2.p2. Source data for each figure are provided with this manuscript in the Source Data file. All code is deposited in the following Bitbucket repository: https://bitbucket.org/kleinstein/impacc-public-code/src/master/azithromycin_manuscript/ .

# Research involving human participants, their data, or biological material

Policy information about studies with human participants or human data. See also policy information about sex, gender (identity/presentation), and sexual orientation and race, ethnicity and racism.

| | |
|---|---|
| Reporting on sex and gender | Sex at birth is reported for each study participant. |
| Reporting on race, ethnicity, or other socially relevant groupings | Self-identified race is reported for each patient, and includes the following groups: Asian, Black, White, Multiple, Other, American Indian/Alaska Native and Native Hawaiian/Pacific Islander.<br><br>Self-identified hispanic ethnicity is reported for each patient. |
| Population characteristics | A comprehensive table of population characteristics is provided in Supplemental Table 1. Briefly, the median age in the cohort was 59.0 (20.0). 61% were male. All participants were hospitalized for COVID-19. 474 participants received no antibiotics, 366 participants received azithromycin and 324 participants received other antibiotics. |
| Recruitment | IMPACC is a prospective longitudinal study that enrolled 1164 patients hospitalized for COVID-1923–25,38,39 from 20 hospitals across 15 academic biomedical centers within the U.S. between May 2020 and March 2021. Inclusion criteria included: 1) age > 18 years; 2) confirmed SARS-CoV-2 positivity by reverse transcription PCR (RT-PCR) testing; and 3) confirmed understanding by participant and/or surrogate of the data to be collected and the study procedures, and willingness to participate in the surveillance cohort. Exclusion criteria included: 1) underlying medical problems which, in the opinion of the investigator, may have been associated with mortality unrelated to COVID-19 within 48 hours of hospitalization; 2) a decision by the patient or surrogate prior to hospitalization to limit care to comfort measures; or 3) medical problems or conditions such as pregnancy which might impact interpretation of the immunologic data obtained. The Department of Health and Human Services Office for Human Research Protections (OHRP) and NIAID concurred that the IMPACC study qualified for public health surveillance exemption. The study protocol was reviewed by each site's institutional review board (IRB), with twelve sites conducting as a public health surveillance study, and three sites integrating the IMPACC study into IRB-approved protocols (The University of Texas at Austin, IRB 2020-04-0117; University of California San Francisco, IRB 20-30497; Case Western Reserve University, IRB STUDY20200573) with participants providing informed consent. Participants enrolled at sites operating as a public health surveillance study were provided information sheets describing the study including the samples to be collected and plans for analysis and data de-identification. Participants who requested not to participate after review of the study plan and information were not enrolled. Participants were not compensated while hospitalized but were subsequently compensated for outpatient visits and surveys. This study was registered at clinicaltrials.gov (NCT04378777) and followed the Strengthening the Reporting of Observational Studies in Epidemiology (STROBE) guidelines. |
| Ethics oversight | The Department of Health and Human Services Office for Human Research Protections (OHRP) and NIAID concurred that the IMPACC study qualified for public health surveillance exemption. The study protocol was reviewed by each site's institutional review board (IRB), with twelve sites conducting as a public health surveillance study, and three sites integrating the IMPACC study into IRB-approved protocols (The University of Texas at Austin, IRB 2020-04-0117; University of California San Francisco, IRB 20-30497; Case Western Reserve University, IRB STUDY20200573) with participants providing informed consent. Participants enrolled at sites operating as a public health surveillance study were provided information sheets describing the study including the samples to be collected and plans for analysis and data de-identification. Participants who requested not to participate after review of the study plan and information were not enrolled. |

Note that full information on the approval of the study protocol must also be provided in the manuscript.

# Field-specific reporting

Please select the one below that is the best fit for your research. If you are not sure, read the appropriate sections before making your selection.

☒ Life sciences  ☐ Behavioural & social sciences  ☐ Ecological, evolutionary & environmental sciences

For a reference copy of the document with all sections, see nature.com/documents/nr-reporting-summary-flat.pdf

# Life sciences study design

All studies must disclose on these points even when the disclosure is negative.

| | |
|---|---|
| Sample size | No sample size calculation was preformed as we leveraged all available data from the large (N=1164) IMPACC observational cohort. Many microbiome and resistome studies have a much smaller population size, and our sample size was robust enough to identify several significant findings. |
| Data exclusions | Data from every patient was analyzed. Negative control water samples were processed in parallel with the participant samples, and we used previously described negative binomial model to exclude microbes and antimicrobial resistance genes likely to be contaminants from the laboratory environment. Based on the results of this model, microbial taxa and antimicrobial resistance genes that were present at a significantly higher abundance in participant samples than in negative controls (FDR < 0.1) were retained for downstream analyses. |
| Replication | No external metatranscriptomic dataset with azithromycin exposure data exists and is available for replication, to our knowledge. |
| Randomization | This was an observational study and thus no randomization was carried out. |
| Blinding | This was an observational study and thus blinding to intervention group was carried out. |

# Reporting for specific materials, systems and methods

We require information from authors about some types of materials, experimental systems and methods used in many studies. Here, indicate whether each material, system or method listed is relevant to your study. If you are not sure if a list item applies to your research, read the appropriate section before selecting a response.

## Materials & experimental systems

| n/a | Involved in the study |
|---|---|
| ☒ | Antibodies |
| ☒ | Eukaryotic cell lines |
| ☒ | Palaeontology and archaeology |
| ☒ | Animals and other organisms |
| ☒ | Clinical data |
| ☒ | Dual use research of concern |
| ☒ | Plants |

## Methods

| n/a | Involved in the study |
|---|---|
| ☒ | ChIP-seq |
| ☒ | Flow cytometry |
| ☒ | MRI-based neuroimaging |

## Plants

| | |
|---|---|
| Seed stocks | *Report on the source of all seed stocks or other plant material used. If applicable, state the seed stock centre and catalogue number. If plant specimens were collected from the field, describe the collection location, date and sampling procedures.* |
| Novel plant genotypes | *Describe the methods by which all novel plant genotypes were produced. This includes those generated by transgenic approaches, gene editing, chemical/radiation-based mutagenesis and hybridization. For transgenic lines, describe the transformation method, the number of independent lines analyzed and the generation upon which experiments were performed. For gene-edited lines, describe the editor used, the endogenous sequence targeted for editing, the targeting guide RNA sequence (if applicable) and how the editor was applied.* |
| Authentication | *Describe any authentication procedures for each seed stock used or novel genotype generated. Describe any experiments used to assess the effect of a mutation and, where applicable, how potential secondary effects (e.g. second site T-DNA insertions, mosiacism, off-target gene editing) were examined.* |

