## [Peer Review File · Nature Microbiology]

Empiric azithromycin alters the upper respiratory microbiome and resistome without anti-inflammatory benefit in COVID-19

Corresponding Author: Dr Charles Langelier

Version 0:

Reviewer comments:

Reviewer #1

(Remarks to the Author)

The authors have responded to my comments and it seems that those are all well addressed.

(Remarks on code availability)

Reviewer #2

(Remarks to the Author)

The authors have addressed all my initial concerns and suggestions. No further changes from my end.

(Remarks on code availability)

Reviewer #3

(Remarks to the Author)

The authors have answered most of my comments except for one major point and one minor point.

The major point concern the rationale behind limiting analysis to 2 million reads per sample which is still, to my point of view, not justified or at least limit the conclusion because 2 Mio reads, even after cleaning, is quite low to explore the full diversity of the microbiome. I would appreciate it if the authors could provide a rarefaction curve showing that with 2 Mio reads, they are reaching the plateau. My argument is also strengthened by the fact that the MetaPhlan provided showed that only 159/458 (34%) of the samples yielded a MetaPhlan4 taxonomic profile, which then indicates that the genome coverage is also quite low.

The minor comment concerns the PERMANOVA analysis.

The author claims that PERMANOVA does not give an R^2 or effect size value. However, PERMANOVA calculates a pseudo-F statistic to test for differences between groups, which then represents the ratio of the between-group sum of squares to the within-group sum of squares (or mean squares). Furthermore, several PERMANOVA test, such as the ADONIS test, provides an R^2 value that measures how much of the total variation in the data is explained by the grouping factor.

(Remarks on code availability)

Decision Letter:

25th November 2025

Dear Dr Langelier,

Thank you for your patience while your manuscript "Empiric Azithromycin in COVID-19 Impacts the Respiratory Microbiome and Antimicrobial Resistome without Anti-inflammatory Benefit" was under peer-review at Nature Microbiology. It has now been seen by the original three referees, whose comments you will find at the of this email. You will see from their comments below that they find your work of interest but a little issue remains. We are very interested in the possibility of publishing your study in

Nature Microbiology, but would like to consider your response to these concerns in the form of a revised manuscript before we make a final decision on publication.

In particular, you will see that R3 is still not convinced with the coverage and the rationale behind using 2 million reads per sample. We recommend that you provide a better justification, hopefully, in the form of a rarefaction curve, as suggested by the referee and address the concern regarding PERMANOVA analysis. The rest referees' reports are clear and the remaining issues should be straightforward to address.

If you have not done so already please begin to revise your manuscript so that it conforms to our Article format instructions at <http://www.nature.com/nmicrobiol/info/final-submission/>

The usual length limit for a Nature Microbiology Article is six display items (figures or tables) and 3,000 words. We have some flexibility, and can allow a revised manuscript at 3,500 words, but please consider this a firm upper limit. There is a trade-off of ~250 words per display item, so if you need more space, you could move a Figure or Table to Supplementary Information.

Some reduction could be achieved by focusing any introductory material and moving it to the start of your opening 'bold' paragraph, whose function is to outline the background to your work, describe in a sentence your new observations, and explain your main conclusions. The discussion should also be limited. Methods should be described in a separate section following the discussion, we do not place a word limit on Methods.

Nature Microbiology titles should give a sense of the main new findings of a manuscript, and should not contain punctuation. Please keep in mind that we strongly discourage active verbs in titles, and that they should ideally fit within 90 characters each (including spaces).

Please include a data availability statement as a separate section after Methods but before references, under the heading "Data Availability". This section should inform readers about the availability of the data used to support the conclusions of your study. This information includes accession codes to public repositories (data banks for protein, DNA or RNA sequences, microarray, proteomics data etc...), references to source data published alongside the paper, unique identifiers such as URLs to data repository entries, or data set DOIs, and any other statement about data availability. At a minimum, you should include the following statement: "The data that support the findings of this study are available from the corresponding author upon request", mentioning any restrictions on availability. If DOIs are provided, we also strongly encourage including these in the Reference list (authors, title, publisher (repository name), identifier, year). For more guidance on how to write this section please see: <http://www.nature.com/authors/policies/data/data-availability-statements-data-citations.pdf>

To improve the accessibility of your paper to readers from other research areas, please pay particular attention to the wording of the paper's opening bold paragraph, which serves both as an introduction and as a brief, non-technical summary in about 150 words. If, however, you require one or two extra sentences to explain your work clearly, please include them even if the paragraph is over-length as a result. The opening paragraph should not contain references. Because scientists from other sub-disciplines will be interested in your results and their implications, it is important to explain essential but specialised terms concisely. We suggest you show your summary paragraph to colleagues in other fields to uncover any problematic concepts.

If your paper is accepted for publication, we will edit your display items electronically so they conform to our house style and will reproduce clearly in print. If necessary, we will re-size figures to fit single or double column width. If your figures contain several parts, the parts should form a neat rectangle when assembled. Choosing the right electronic format at this stage will speed up the processing of your paper and give the best possible results in print. We would like the figures to be supplied as vector files - EPS, PDF, AI or postscript (PS) file formats (not raster or bitmap files), preferably generated with vector-graphics software (Adobe Illustrator for example). Please try to ensure that all figures are non-flattened and fully editable. All images should be at least 300 dpi resolution (when figures are scaled to approximately the size that they are to be printed at) and in RGB colour format. Please do not submit Jpeg or flattened TIFF files. Please see also 'Guidelines for Electronic Submission of Figures' at the end of this letter for further detail.

Figure legends must provide a brief description of the figure and the symbols used, within 350 words, including definitions of any error bars employed in the figures.

When submitting the revised version of your manuscript, please pay close attention to our [href="https://www.nature.com/nature-research/editorial-policies/image-integrity">Digital Image Integrity Guidelines](https://www.nature.com/nature-research/editorial-policies/image-integrity) and to the following points below:

EXTENDED DATA FIGURES

Please include a statement before the acknowledgements naming the author to whom correspondence and requests for materials should be addressed.

Finally, we require authors to include a statement of their individual contributions to the paper -- such as experimental work, project planning, data analysis, etc. -- immediately after the acknowledgements. The statement should be short, and refer to authors by their initials. For details please see the Authorship section of our joint Editorial policies at http://www.nature.com/authors/editorial_policies/authorship.html

* include a point-by-point response to any editorial suggestions and to our referees. Please include your response to the editorial suggestions in your cover letter, and please upload your response to the referees as a separate document.

* ensure it complies with our format requirements for Letters as set out in our guide to authors at www.nature.com/nmicrobiol/info/gta/

* state in a cover note the length of the text, methods and legends; the number of references; number and estimated final size of figures and tables

* resubmit electronically if possible using the link below to access your home page:

Link Redacted

*This url links to your confidential homepage and associated information about manuscripts you may have submitted or be reviewing for us. If you wish to forward this e-mail to co-authors, please delete this link to your homepage first.

Please ensure that all correspondence is marked with your Nature Microbiology reference number in the subject line.

Nature Microbiology is committed to improving transparency in authorship. As part of our efforts in this direction, we are now requesting that all authors identified as 'corresponding author' on published papers create and link their Open Researcher and Contributor Identifier (ORCID) with their account on the Manuscript Tracking System (MTS), prior to acceptance. This applies to primary research papers only. ORCID helps the scientific community achieve unambiguous attribution of all scholarly contributions. You can create and link your ORCID from the home page of the MTS by clicking on 'Modify my Springer Nature account'. For more information please visit www.springernature.com/orcid

We hope to receive your revised paper within three weeks. If you cannot send it within this time, please let us know.

Yours sincerely,

Reviewers Comments:

Reviewer #1 (Remarks to the Author):

The authors have responded to my comments and it seems that those are all well addressed.

Reviewer #2 (Remarks to the Author):

The authors have addressed all my initial concerns and suggestions. No further changes from my end.

Reviewer #3 (Remarks to the Author):

The authors have answered most of my comments except for one major point and one minor point.

The major point concern the rationale behind limiting analysis to 2 million reads per sample which is still, to my point of view, not justified or at least limit the conclusion because 2 Mio reads, even after cleaning, is quite low to explore the full diversity of the microbiome. I would appreciate it if the authors could provide a rarefaction curve showing that with 2 Mio reads, they are reaching the plateau. My argument is also strengthened by the fact that the Metaphlan provided showed that only 159/458 (34%) of the samples yielded a MetaPhlan4 taxonomic profile, which then indicates that the genome coverage is also quite low.

The minor comment concerns the PERMANOVA analysis.

The author claims that PERMANOVA does not give an R^2 or effect size value. However, PERMANOVA calculates a pseudo-F statistic to test for differences between groups, which then represents the ratio of the between-group sum of squares to the within-group sum of squares (or mean squares). Furthermore, several PERMANOVA test, such as the ADONIS test, provides an R^2 value that measures how much of the total variation in the data is explained by the grouping factor.

Version 1:

Reviewer comments:

Reviewer #3

(Remarks to the Author)

The authors have responded to all of my comments. I do not have any more remarks and I thank the authors for their patience and dedication.

(Remarks on code availability)

Decision Letter:

Our ref: NMICROBIOL-25082844A

16th December 2025

Dear Dr. Langelier,

Thank you for submitting your revised manuscript "Empiric Azithromycin in COVID-19 Impacts the Respiratory Microbiome and Antimicrobial Resistome without Anti-inflammatory Benefit" (NMICROBIOL-25082844A). It has now been seen by the original referees and their comments are below. In light of those comments, we are happy to inform that we would be publish it in principle, in Nature Microbiology, pending minor revisions to satisfy the referees' final requests and to comply with our editorial and formatting guidelines.

Thank you again for your interest in Nature Microbiology Please do not hesitate to contact me if you have any questions.

Happy holidays!

Reviewer #3 (Remarks to the Author):

The authors have responded to all of my comments. I do not have any more remarks and I thank the authors for their patience and dedication.

Version 2:

Decision Letter:

4th February 2026

Dear Dr Langelier,

I am pleased to accept your Article "Empiric azithromycin alters the upper respiratory microbiome and resistome without anti-inflammatory benefit in COVID-19" for publication in Nature Microbiology. Thank you for having chosen to submit your work to us and many congratulations.

Authors may need to take specific actions to achieve compliance with funder and institutional open access mandates. If your research is supported by a funder that requires immediate open access (e.g. according to [a href="https://www.springernature.com/gp/open-science/plan-s-compliance"> Plan S principles](https://www.springernature.com/gp/open-science/plan-s-compliance) or the [a href="https://www.springernature.com/gp/open-science/us-federal-agency-compliance"> NIH public access policy](https://www.springernature.com/gp/open-science/us-federal-agency-compliance)) then you should select the gold OA route, and we will direct you to the compliant route where possible. Because authors warrant under our subscription licensing terms that they haven't committed to licensing any version of their article under a licence inconsistent with the terms of our agreement – including the applicable embargo period – publication under the subscription model isn't suitable for authors whose funders require no embargo.

An online order form for reprints of your paper is available at [a href="https://www.nature.com/reprints/author-reprints.html">https://www.nature.com/reprints/author-reprints.html](https://www.nature.com/reprints/author-reprints.html). All co-authors, authors' institutions and authors' funding agencies can order reprints using the form appropriate to their geographical region.

With kind regards,

P.S. Click on the following link if you would like to recommend Nature Microbiology to your librarian
<http://www.nature.com/subscriptions/recommend.html#forms>

** Visit the Springer Nature Editorial and Publishing website at http://editorial-jobs.springernature.com?utm_source=ejP_NMicro_email&utm_medium=ejP_NMicro_email&utm_campaign=ejP_NMicro for more information about our career opportunities. If you have any questions please click [here](mailto:editorial.publishing.jobs@springernature.com).

Referee expertise:

Referee #1: pulmonary clinician, respiratory microbiome

Referee #2: respiratory microbiome and medicine, AMR

Referee #3: microbiome analyses

Reviewers' Comments:

Reviewer #1 (Remarks to the Author):

In this paper, Glascock et al., investigated the nasal metatranscriptome among patients treated with azithromycin (n=366, 31.4%) vs those who received no antibiotics (n=474, 40.7%) or antibiotics other than azithromycin (n=324, 27.8%). Samples were obtained longitudinally. Their data showed that AZM use was associated with increase of fungal relative abundance and potentially pathogenic taxa such as *Klebsiella* and *Staphylococcus*. It was also associated with increased of macrolide/lincosamide/streptogramin (MLS) in the resistome, which was associated with the abundance of several taxa, including both commensal (e.g., *Dolosigranulum*, *Corynebacterium*) and potentially pathogenic genera (e.g., *Streptococcus*, *Staphylococcus*). No differences were noted in inflammatory markers based on host transcriptome from blood or nasal samples.

The authors have made interesting observations, being measured in their interpretation of their results, and addressing their limitation. Overall, several of the observations were expected. However, one interesting aspect to highlight is how rapid azithromycin administration seem to effect the upper airway resistome, how much of those changes may occur among taxa not readily recognized as respiratory pathogens and how little (actually none) changes were seen in the host transcriptome. This latter observation is perhaps the most provocative given how intricated is the idea that macrolides have a lot of anti-inflammatory activity.

The major issue is that the observational nature of their design prevents us from having a definitive interpretation of their results. Overall, I don't have many comments, just a few for clarification.

Below are my point-by-point comments

Major comments

1. The authors identified a correlation of MLS genes and both potentially pathogenic (e.g., *Staphylococcus*, *Streptococcus*) as well as common commensal (e.g., *Corynebacterium*, *Dolosigranulum*) genera. While informative, it would be also be informative to try to determine taxonomic origin of those MLS genes. This could be done via blast, assembly of contigs or other analytical approaches. In particular, I think the point that MLS genes can be contained within bacteria not traditionally thought to be respiratory pathogens is an important observation.

We appreciate this suggestion and agree that identifying MLS genes within bacteria not traditionally thought to be respiratory pathogens is an important observation. To address this feedback, we have taken the following approaches to predict species of origin for the MLS genes detected:

1. Performing species level multidimensional correlation analyses, similar to the genus-level analyses described in Fig. 4b. We have included results in a new Supplementary Table 4 and have highlighted those species within significant differentially abundant taxa in a new Supplementary Fig. 7.

Supplementary Figure 7. Species-level correlations within the airway resistome and microbiome. Species-level Spearman correlation analysis among differentially abundant genera from Fig. 2f. Only taxa with significant correlations are displayed. Color bar reflects Spearman correlation coefficient. *** $p_{adj} < 0.001$; ** $p_{adj} < 0.01$; * $p_{adj} < 0.05$.

2. Employing the Comprehensive Antibiotic Resistance Gene Database pathogen-of-origin tool²⁹ to match k-mers from detected MLS resistance gene sequences against species present in the tool's reference database, which consists of 437 bacterial genomes, most of which represent established pathogens.

Supplementary Figure 8. Species-level taxonomic assignment of MLS resistance genes based on CARD pathogen-of-origin assignments. Heatmap of CARD pathogen-of-origin assignments and MLS ARGs. Red cells denote the assignment of a given ARG to a given taxon by CARD. Genomic location information in parentheses.

Together, these species-level analyses support and expand upon the relationships identified from our genus-level multi-dimensional correlation analyses. For instance, the correlation between *Staphylococcus aureus* and the MLS resistance genes *mphC* and *msrA*; and between *Corynebacterium segmentosum* and *ermX*. Application of the CARD RGI pathogen-of-origin tool also predicted additional linkages, such as those between *ermC* and both *S. aureus* and *S. epidermidis*.

We now describe the results of these additional analyses in a new Supplementary Fig. 7 (species level correlations) and Supplementary Figure 8 (CARD pathogen-of-origin k-mer matching), and in the text as follows:

Line 243: Relationships were further reinforced with species-level multi-dimensional correlation analyses (e.g., *Staphylococcus aureus* and *mphC*, *msrA*) (**Supplementary Fig. 7, Supplementary Table 4**). Application of the CARD RGI pathogen-of-origin tool³¹, which matches k-mers from detected ARG sequences against a database of established bacterial pathogens, provided additional insight regarding linkages between macrolide resistance genes and specific species (**Supplementary Fig. 8**). For instance, this additional analysis highlighted a relationship between *Streptococcus pyogenes* and *ermA*, and additional linkages between *ermC* and both *Staphylococcus aureus* and *Staphylococcus epidermidis*.

Minor comments

1. Figure 2E shows negative values at day 0 for Bray Curtis. Not sure how the authors got negative values. Please explain.

We thank the reviewer for pointing out this issue, which occurred due to model fitting with limited day zero data. We have addressed this by revising the approach in Figure 2e to both solve this problem and better align with the point of the text. The main update is a change to the X axis with the model now serving as a function of days from the first sample. Previously, the graph showed distance from the first collected sample, but as a function of days from hospitalization, which did not best account for the varying day of first sample. As a result, this revised version is simplified to directly model against days from first sample and better demonstrates the relationship between antibiotic exposure and the dynamics of respiratory microbiome community composition.

2. For fig 2F, Heatmap highlighting differentially abundant genera in patients treated with azithromycin for 5 ± 1 days compared to those who received no antibiotics or other antibiotics. Not sure how this was calculated, unless the group who received no antibiotics were grouped with the ones that received other antibiotics and this combined group was compared with those that did receive azithromycin.

We appreciate the opportunity to clarify that in this analysis, patients who received azithromycin were compared against all others grouped together (i.e., those who received no antibiotics and those who received other antibiotics). We have now added the following to the figure legend to better clarify the comparison groups:

Fig. 2 Legend: Heatmap highlighting differentially abundant genera in patients treated with azithromycin for 5 ± 1 days compared to other patients (No-Abx + Other-Abx groups).

3. Line 190: To understand the impact of azithromycin exposure on the respiratory antimicrobial resistome, we first compared Shannon Diversity Index across groups. This was probably done in ARG table, no? this should be clear to avoid confusing with taxonomic tables.

Yes, the reviewer is correct, and we appreciate them highlighting the need to clarify this point. We have now clarified this better in the text as follows:

Line 193: To understand the impact of azithromycin exposure on the respiratory antimicrobial resistome, we first compared ARG alpha diversity across groups. Azithromycin exposure was associated with an increase in the resistome Shannon Diversity Index...

Reviewer #1 (Remarks on code availability): I reviewed briefly without running it and it seems pretty clean.

We appreciate the feedback on our code.

Reviewer #2 (Remarks to the Author):

This submitted manuscript looks at the prospective, multi-centre IMPACC cohort (n = 1,164 adults hospitalised with COVID-19 in the USA) to quantify the ecological and functional impact of empiric azithromycin therapy on the upper-airway microbiome, resistome and host transcriptome. Longitudinal nasal metatranscriptomics showed that even short courses of azithromycin (median 2 days) precipitated a rapid fall in bacterial abundance, a rise in fungal burden and an enrichment of potentially pathogenic genera (e.g., *Klebsiella*, *Staphylococcus*). Concomitantly, the number and proportional representation of macrolide/lincosamide/streptogramin (MLS) resistance genes, particularly *ermC*, *msrA* and *ermX* increased within 24 h and remained elevated for >7 days after discontinuation. No change in SARS-CoV-2 load or inflammatory gene expression was detected, undermining any putative anti-inflammatory benefit. Collectively, the study provides timely evidence that indiscriminate azithromycin use during viral respiratory illness can accelerate airway dysbiosis and amplify functional antimicrobial resistance. The study addresses an urgent clinical question with state-of-the-art methodology, but stronger control for confounding, clearer presentation of quantitative results and refinement of causal language are required before the work can meet Nature Medicine's standards for causal inference and translational impact and suggest the following:

Abstract

-Omits absolute effect sizes (e.g., fold-change in MLS genes) and confidence intervals, should include

We thank the reviewer for this feedback and have now included fold change and 95% confidence intervals in the abstract and text. We would like to note that these values are also included in the source data file.

-The concluding sentence implies causality (leads to dysbiosis) despite the observational design, would revise.

We appreciate this point and have now revised the wording as follows to avoid implying causality:

Line 64: Taken together, our findings demonstrate that azithromycin treatment in COVID-19 associates with dysbiosis of the upper respiratory microbiome and changes in the expression of MLS resistance genes, without apparent anti-inflammatory benefit.

-Limitations (upper-airway sampling, confounding by indication) are not acknowledged and perhaps should be included if room.

We appreciate this point however to accommodate the format for Nature Microbiology we needed to reduce the length of our abstract by > 150 words and thus have only included limitations in the main manuscript text. If the reviewer or editor feels strongly about this point, we are happy to revise the abstract further to highlight limitations.

Introduction

-Provides good epidemiologic context but could be more concise; the first three paragraphs repeat global AMR statistics without directly motivating the current study, would be revised.

We appreciate this feedback and have now revised the introduction by removing the first three paragraphs as suggested.

-Novelty is well stated (lack of metatranscriptomic data in acute infections), yet the rationale for choosing nasal swabs over lower-airway specimens is not justified and needs to be stated.

We appreciate this feedback and the opportunity to clarify our rationale. We elected to analyze nasal swabs because they were available on nearly all patients in the cohort, spanning the full range of disease severity among hospitalized patients. In contrast, lower airway tracheal aspirate specimens would only be obtainable from critically ill patients requiring mechanical ventilation, who represented < 30% of the cohort. Bronchoscopy was rarely performed at most study site hospitals due to strict infection control precautions during the study period with respect to aerosol generating procedures. Furthermore, bronchoscopy is not as regularly performed in patients requiring mechanical ventilation due to safety risks. Finally, many patients are unable to produce sputum without saline induction, another aerosol generating procedure, and sputum collection was not routinely performed at study site hospitals. We have now clarified these reasons for studying nasal swabs in the text as follows:

Methods, Line 380: We elected to analyze nasal swabs because they were available on nearly all patients in the cohort, spanning the full range of disease severity. In contrast, lower airway tracheal aspirate specimens would only be obtainable from critically ill patients requiring mechanical ventilation, who represented < 30% of the cohort. Bronchoscopy was rarely performed at most study site hospitals due to strict infection control precautions during the study period with respect to aerosol generating procedures. Finally, many patients are unable to produce sputum without saline induction, another aerosol generating procedure, and sputum collection was not routinely performed at study site hospitals.

-A brief synopsis of previous negative RCTs of azithromycin in COVID-19 would strengthen the clinical relevance and should be included for context.

We appreciate this suggestion and have added the following:

Line 76: Randomized controlled trials, however, subsequently demonstrated that azithromycin conferred no clinical benefit in the treatment of COVID-19^{11,12}. These included hospitalized patients (RECOVERY trial¹³, n=7763), hospitalized patients with severe COVID-19 (COALITION II trial¹⁴, n=397), mild-moderate COVID-19 (ATOMIC-II¹², n=298), and outpatients (PRINCIPLE¹⁵, n=1323).

Methods

- The multi-omic pipeline with background subtraction is a strength, but antibiotic exposure is classified only at two static windows (1 ± 1 day, 5 ± 1 days); intermediate dynamics are modelled yet not validated with sensitivity analyses and should be considered.

First, we thank the reviewer for highlighting the strength of our bioinformatics pipeline. With respect to the timepoints analyzed: the typical course of azithromycin treatment is five days, and thus our day 5 ± 1 timepoint selection was motivated by the clinical relevance of this timeframe. The day 1 ± 1 timepoint was then selected to capture the earliest available sample following antibiotic exposure. We considered a third intermediate day 3 ± 1 timepoint, however that intermediate timepoint would have had overlapping sample windows and thus comparative statistics at that timepoint would have been problematic. We would also like to emphasize that our generalized additive mixed models (GAMM) demonstrated no outliers on day 3, and also note that the intermediate timepoints are represented in the GAMM data presented in Figs. 3d-e and Supplementary Fig. 2. We have now clarified our reasoning in the manuscript as follows:

Line 459: The day 5 ± 1 timepoint was chosen for clinical relevance based on the typical azithromycin treatment course of five days. The day 1 ± 1 timepoint was selected to capture the earliest available sample following antibiotic exposure. Given that a third intermediate timepoint at day 3 ± 1 would have included overlapping samples, we instead assessed these intermediate timepoints through GAMM analysis.

-Severity groups differed between antibiotic strata; although trajectory group is a covariate, residual confounding is likely. Consider propensity-score or inverse-probability-weighting to include to account for this.

We appreciate this point, and as the reviewer indicates, we are already accounting for COVID-19 severity by incorporating trajectory group (TG) as a covariate in our models. We would like to respectfully note that residual confounding still occurs in propensity score matching or inverse probability weighting. Nonetheless, to address this input, we have carried out a sensitivity analysis matching patients based on their disease severity trajectory group (moderate TG 1-3; or severe TG 4-5). Importantly, we found that our primary conclusions remained unchanged in this sensitivity analysis. More specifically, we observed that the longitudinal dynamics of MLS ARG richness and MLS percent of resistome did not differ by trajectory group. In both analyses, the 95% confidence intervals completely overlapped and there was no significant difference between the models based upon disease severity with respect to trajectory group ($p=0.86$ and $p=0.54$), respectfully). These analyses have been incorporated into our manuscript as Supplementary Figs. 6a and 6b.

Supplemental Figure 6. Sensitivity analyses of MLS resistance gene longitudinal dynamics stratified by COVID-19 severity trajectory group (TG). **a** Generalized additive mixed model (GAMM) demonstrating changes in the number of detectably expressed MLS resistance genes over time stratified by COVID-19 severity (TG 1-3, red, mild/moderate disease) versus (TG 4-5, blue, severe/fatal disease). **b** Longitudinal changes in the proportional representation of MLS resistance genes (red) in the nasal resistome over time stratified by COVID-19 severity (TG 1-3, red, mild/moderate disease) versus (TG 4-5, blue, severe/fatal disease). The smoothed curves represent the estimated non-linear relationship between variables. The center line represents the predicted value, and the shaded area denotes the 95% confidence interval. Global p-values were generated from a likelihood-ratio test comparing a non-stratified GAMM to a full GAMM with group-specific trajectories.

-The negative-control strategy is well described, but batch-effect correction across four sequencing phases could be summarized in a flow-diagram for transparency and clarity.

We appreciate this suggestion and have now included a flow diagram to improve transparency and clarity, as a new Supplementary Fig. 1.

Upper-airway focus limits inference about the lung; justify why BAL or sputum samples were not collected.

We agree with the reviewer and recognize this as a limitation, which we have further highlighted in the discussion. As noted above, we elected to analyze nasal swabs because they were available on nearly all patients in the cohort, spanning the full range of disease severity among hospitalized patients. In contrast, lower airway tracheal aspirate specimens would only be obtainable from critically ill patients requiring mechanical ventilation, who represented < 30% of the cohort. Bronchoscopy was rarely performed at most study site hospitals due to strict infection control precautions during the study period with respect to aerosol generating procedures. Finally, many patients are unable to produce sputum without saline induction, another aerosol generating procedure, and sputum collection was not routinely performed at study site hospitals. We have now clarified these reasons for studying nasal swabs in the text as follows:

Methods, Line 380: We elected to analyze nasal swabs because they were available on nearly all patients in the cohort, spanning the full range of disease severity. In contrast, lower airway tracheal aspirate specimens would only be obtainable from critically ill patients requiring mechanical ventilation, who represented < 30% of the cohort. Bronchoscopy was rarely performed at most study site hospitals due to strict infection control precautions during the study period with respect to aerosol generating procedures. Finally, many patients are unable to produce sputum without saline induction, another aerosol generating procedure, and sputum collection was not routinely performed at study site hospitals.

Discussion (limitations section), Line 332: In addition, our analyses of nasal swabs, a widely available specimen type from patients hospitalized for COVID-19, were limited to the upper airway and thus may not reflect microbial changes occurring in the lungs.

-Ethics section is comprehensive, yet inclusion of unvaccinated patients only (May 2020–Mar 2021) should be highlighted as a temporal limitation as another confounder.

Discussion (limitations section), Line 337: Because our study preceded widespread COVID-19 vaccination, results were not influenced by vaccination status but could potentially differ in a contemporary vaccinated cohort.

Results

-Important quantitative details (median log-change in bacterial RPM, absolute Δ MLS-ARG proportion) are confined to figures; summarize in text for readers.

We appreciate this suggestion and have now added fold change and 95% confidence intervals throughout the manuscript, in addition to the source data file.

-Statistical models adjust for six co-administered antibiotics, but effect estimates for those agents are not shown—reporting them would clarify specificity to azithromycin.

We thank the reviewer for this suggestion we have now included the information in the source data. Weights for each covariate and p values are included as well.

-Persistence of MLS genes 7–10 days post-therapy is a key message but not well shown, so providing individual-level trajectories (e.g., spaghetti plot) would illustrate heterogeneity.

We appreciate this suggestion and have added new Supplementary Figs. 5b and 5c demonstrating individual-level trajectories.

Supplementary Figure 5. MLS ARG class metrics show no significant decrease after cessation of azithromycin. **a.** MLS class percent of resistome before and after azithromycin cessation. P values are based on a linear mixed-effects model. Boxplots display the median, first and third quartile, and range. Data points are randomly jittered along the x-axis for visual clarity. **b.** Spaghetti plot of MLS percent of resistome over time following azithromycin cessation. **c.** Spaghetti plot of MLS ARG richness over time following azithromycin cessation. Colors represent different participants. Repeat measures from the same participant are connected.

-The null finding for host inflammatory gene expression is potentially under-powered, perhaps can indicate sample size for PBMC and nasal transcriptomics.

Response. We thank the reviewer for this suggestion. We've now clarified that the number of samples was 238 and 217 for the PBMC and nasal host transcriptomics analyses, respectively. We would also like to note that this represents a larger sample size than many recent high profile transcriptional profiling manuscripts (e.g., Anderson et al. 2023. Nature Medicine; Kalantar et al. 2022. Nature Microbiology; Viz-Lasheras et al. 2025. Nature Communications).

Line 551: ...together leaving 217 nasal and 238 PBMC samples available for analysis.

Figures

-Colour palette for antibiotic groups (blue/green/orange) is intuitive, but overlapping shades for early vs late azithromycin (orange/red) hinder discrimination (Fig. 2d) and should be revised.

We appreciate this input and have revised the colors to improve discrimination between groups:

-Figure 1d lists frequency of co-prescribed antibiotics but lacks denominators; re-express as percentage of azithromycin recipients.

We thank the reviewer for this suggestion and have now included denominators and have expressed as percentage of azithromycin recipients.

-Statistical annotations (**) are provided, yet actual P values or effect sizes would be more informative to include.

We appreciate this input and have now included the p values in the source data file. Because all adjusted p values were $< 1e-9$, we did not annotate each box with the value. We have also noted this in the figure legend.

-Network plot (Fig. 4c) is visually dense; consider interactive supplement or pruning to strongest correlations to be more clear.

We appreciate this suggestion and have pruned the network plot to retain just those with the most significant correlations ($\rho > 0.2$ and $p_{adj} < 0.05$).

C

-While useful for robustness checks, but Supplementary Fig. 2 (GAMM of bacterial/fungal RPM over time) has a main conclusion and merits inclusion in the primary results.

We thank the reviewer for the suggestion but would like to respectfully note that the GAMM analyses did not demonstrate any statistically significant ($p_{adj} < 0.05$) associations between azithromycin exposure and bacterial or fungal RPM. While the trends are interesting, they do not reflect a main conclusion and thus we have elected to retain them in the supplement.

-Supplementary Fig. 4 duplicates information in Fig. 3f; consolidate to avoid redundancy.

We would like to clarify that these figures represent different metrics. Fig 3f is plotting richness (number of detectably expressed MLS resistance genes) while Supplementary Fig. 4 is plotting the percent of the airway resistome represented by MLS resistance genes.

-Supplementary tables lack a detailed data dictionary, thus suggest provide variable definitions and units for ease of reuse.

We appreciate this suggestion and have now added a data dictionary to each supplementary table and better clarified abbreviations in the legends.

Discussion

-Balanced appraisal of strengths and limitations, yet the causal language (azithromycin leads to) should be tempered given observational design and is overstating.

We appreciate this point and have modified this sentence as follows:

Line 289: Importantly, we find that azithromycin exposure correlates not only with an increase in the potential for resistance within the microbiome, but also with the functional expression of MLS resistance genes.

-Comparison with gut-focused MORDOR data is insightful, but mechanistic speculation (age-related ARG reservoir) could be shortened given lack of data.

We appreciate this suggestion and have shortened this mechanistic speculation.

Line 318: One possible explanation may lie in the age and demographic differences of the studied populations. Alternatively, differences may be attributable to sampling of the respiratory versus gut microbiome, or the use of metatranscriptomics versus metagenomic DNA sequencing.

-Future directions (randomised trials with airway and gut sampling) are appropriate but also suggest exploring lower-airway samples and metagenome-assembled genomes (MAGs) to link ARGs to taxa.

We appreciate this point and have added these important additional future directions:

Line 342: Evaluation of lower airway samples and metagenome-assembled genomes could further extend our understanding of antibiotic exposure, including on specific taxa carrying ARGs.

-Consider public-health implications: findings support stewardship efforts to curtail empiric macrolide use in viral infections. Can the authors speculate a role for this?

We agree that our findings have public health implications and support the importance of stewardship efforts surrounding empiric macrolide use in viral infections. We have revised the discussion as follows to emphasize this important point:

Line 347: Taken together, our findings suggest that empiric macrolide use in patients with viral respiratory infections may contribute to antimicrobial resistance and public health risks, underscoring the importance of stewardship efforts.

Reviewer #3 (Remarks to the Author):

A. Summary of the Key Results

The study by Glascock et al. investigates the impact of azithromycin on the respiratory microbiome and inflammation in COVID-19 patients. Employing RNAseq-based profiling, the authors claim that azithromycin exposure is associated with an enrichment of potentially pathogenic taxa such as *Staphylococcus* and *Klebsiella*, and a depletion of commensals such as *Neisseria* and *Fusobacterium*. Furthermore, they report persistent enrichment of macrolide-lincosamide-streptogramin (MLS) resistance genes (ARGs) and significant shifts in beta-diversity.

B. Originality and Significance

The aim of the study is timely and relevant, especially considering the widespread use of azithromycin as empirical therapy during the COVID-19 pandemic. The application of RNAseq to investigate the airway microbiome and resistome represents a novel and potentially valuable methodological advancement.

We appreciate that the reviewer finds our study timely and relevant, and that the application of RNAseq to investigate the airway microbiome and resistome represents a novel and potentially valuable methodological advancement.

C. Data & Methodology

The methods section lacks critical detail that reduce the reproducibility and evaluation of data quality.

1) Library preparation and sequencing: The manuscript does not clarify whether single-end or paired-end sequencing was used. If paired-end 50 bp reads were used, they are often insufficient for confident taxonomic assignment, while single-end reads compromise assembly quality with tools like SPAdes.

We appreciate the reviewer highlighting the need to provide these additional methodological details and would like to clarify that paired end 100 bp sequencing was carried out using a NovaSeq S4 flowcell. We have now clarified this in the Methods section as follows:

Line 405: Libraries, normalized to 10 nM, underwent paired-end 100 base pair sequencing on an Illumina NovaSeq 6000 instrument using S4 flow cells, targeting 50 million reads per sample.

2) Host decontamination based on short reads need a clear explanation on how host RNA was removed or how microbial reads were confidently assigned.

We appreciate this point and have directly addressed the need for a more comprehensive and clear explanation by adding the following to the Methods section:

Line 410: Metatranscriptomic data was processed using the open-source CZ ID (<https://czid.org/>) Illumina mNGS pipeline (v7.1)^{39,40}. Host reads were first removed by subtractive alignment against the GRCh38 human reference genome⁴¹ using STAR⁴². Adapters were then removed using Trimmomatic⁴³, low quality reads filtered using PriceSeq⁴⁴ and low complexity reads filtered using the Lempel-Ziv-Welch algorithm. Duplicate reads were identified and compressed using czid-dedup³⁹. A final scrub of any remaining host reads was carried out by re-alignment to GRCh38 using Bowtie2⁴⁵. After performing these filtering steps, the remaining microbial data were subsampled to 2 million total reads. Taxonomic classification was then carried out by aligning reads to the NCBI nucleotide (NT) database using GSNAP-L⁴⁶. In parallel, short reads were assembled into contiguous sequences using (SPAdes)⁴⁷, which then underwent alignment against accessions from the identified taxa to improve mapping specificity^{39,47}.

3) The authors did not specify whether rRNA was depleted, which is crucial given that rRNA typically dominates total RNA and would complicate contig assembly and gene detection.

We would like to clarify that indeed rRNA was depleted, and have updated the methods section as follows:

Line 398: Extracted RNA then underwent rRNA depletion, cDNA synthesis, and library construction using the Illumina Total Stranded RNA Prep with Ribo-Zero Plus kit, following the manufacturer's instructions, and automated on a Perkin Elmer Sciclone NGSx Workstation. Control samples (Thermo Fisher, cat# AM7832) were also included to assess for and minimize inter-batch variability.

4) The rationale behind limiting analysis to 2 million reads per sample is not justified and may undermine the depth required for comprehensive microbiome and resistome profiling.

We appreciate the opportunity to clarify that the subsampling occurs after all quality filtering and host sequence removal is complete. Given that respiratory samples are comprised primarily of human sequence, most samples (2513/2846, 88.3%) did not require subsampling at all. The subsampling to 2 million paired-end microbial reads is a function of the CZID pipeline, and was originally incorporated to make comparison across samples more appropriate when high variability exists in the sequencing depth, and to increase processing speed for samples with disproportionately high numbers of microbial reads.

We believe that this subsampling approach is appropriate in our study for two reasons. First, the reads from 88.3% of samples were completely captured by the 2 million read cutoff; thus, we are fully profiling the data for the majority of the samples. Second, for the remaining samples that underwent subsampling, the total number of non-host reads passing QC was highly variable,

ranging from just over 2 million to 32 million reads. In cases where some samples are sequenced much more deeply than others, it is common to observe differences in taxon presence and diversity resulting from the variation in sequencing depth, not representative of real biological variation. One approach frequently taken in microbiome studies to address this scenario is rarefying the data to a lower total read count. The subsampling approach implemented in our study allowed us to keep all of the data for the majority of samples while in effect rarefying the data for the very deeply sequenced samples to allow for more statistically sound comparison across samples.

5) Contamination in negative controls is concerning given RNA's instability; ultra-clean conditions (e.g., those described by Pust et al., 2020) should be confirmed. Furthermore, details about the taxa and ARGs found in these controls are missing.

First, we would like to confirm that we used ultra-clean conditions and a rigorous protocol to prevent contamination when processing all samples. This included wearing personal protective equipment including gown, sterile gloves and face shield when working with these samples, using RNaseZap (Ambion) to wipe down pipets and bench surfaces to remove RNase contamination, and additionally spraying with 70% ethanol to disinfect and further clean. In addition, samples were thawed by extraction batch, and all tubes and plates were kept sealed or covered at all times until they were utilized. We have now clarified this in the text as follows:

Line 391: We used a rigorous protocol and an ultra-clean laboratory space to prevent contamination when processing all samples. This involved wearing personal protective equipment including a gown, sterile gloves and a face shield when working with samples, using RNaseZap (Ambion, Inc.) to wipe down pipets and bench surfaces to remove RNase contamination, and additionally spraying with 70% ethanol to disinfect and further clean. In addition, samples were thawed by extraction batch, and all tubes and plates were kept sealed or covered at all times until they were utilized.

In addition, we have now added two new supplemental data files tabulating all reads passing the described QC filters in the 26 negative control water samples mapping to both microbes (Supplementary Table 7) and ARGs (Supplementary Table 8). As noted in the manuscript, these controls were then used to perform background correction using a negative binomial model for both microbial and resistome profiling purposes.

Line 436: Twenty-six negative control samples consisting of double distilled water were processed and sequenced alongside clinical samples in the IMPACC cohort to enable the characterization and subtraction of background contamination. Reads mapping to taxa and ARGs in these control samples retained for downstream analyses after the above-described quality control measures are tabulated in Supplementary Tables 7 and 8, respectively.

6) RNAseq data are not publicly accessible. It is unclear whether they were deposited in SRA or ENA.

We would like to clarify that the raw RNAseq data, which is protected, has been deposited in the NIH NIAID's official ImmPort data repository. Information about the raw data deposition is provided in the Data and code availability section of the methods. As the raw sequencing data are protected, we have included source data for all of the figures in this manuscript as the Source data file, which report all of the results directly plotted in the graphs in effort to reduce this barrier.

D. Appropriate Use of Statistics and Treatment of Uncertainties

Some claims in the manuscript are based on statistical significance without providing effect sizes. For example, while shifts in beta-diversity are statistically significant, R^2 values are missing, which are essential for interpreting biological relevance. The reported enrichment in MLS ARGs is not statistically significant, yet the authors emphasize this point. Transparency in presenting p-values alongside effect sizes is needed to accurately convey the strength and significance of findings.

We appreciate the suggestion to add effect sizes, and have done so throughout the manuscript. We are confused by the reviewer's comment stating that, The reported enrichment in MLS ARGs is not statistically significant... because the significant association between azithromycin exposure and enrichment in MLS ARGs is a central finding of our study. Figure 3c demonstrates that azithromycin exposure is associated with a significant increase in the number of detectably expressed MLS ARGs at each time point examined, with respect to either the No-Abx group or the Other-Abx group. Figure 3d demonstrates that the number of days of azithromycin exposure is significantly associated with the number of detectably expressed MLS ARGs, and figure 3e demonstrates that the number of days of azithromycin exposure is significantly associated with the proportion of the respiratory resistome comprised of MLS ARGs.

In addition, we would like to clarify that we used the PERMANOVA statistical test to evaluate differences in beta diversity. This is the most widely used and accepted test to evaluate beta diversity in the microbiome literature and p-values but not R^2 values are returned by this test.

E. Conclusions: Robustness, Validity, Reliability

Claims around the enrichment of pathogenic taxa (e.g., *Staphylococcus*, *Klebsiella*, *Streptococcus*) are made at the genus level, which is insufficient to identify truly pathogenic species without further taxonomic resolution.

We thank the reviewer for this comment and agree that species-level taxonomic resolution would be preferable in this case. Unfortunately, we have observed that spurious species alignments, typically attributed to ambiguous read mapping, can bias the results. This is especially true in a sparse sample type like a nasal swab. To address this, we have paired our genus-level differential abundance analysis (Fig. 2f) with an additional species-level plot (Fig. 2g) illustrating the relative abundance of the most prevalent species in each of the differentially abundant genera, stratified

by treatment group (azithromycin vs. non-azithromycin). Here the non-azithromycin group includes both No-Abx and Other-Abx samples. This allows the reader to visualize the species level differences within these genera without making any statistical claims at the species level that could be biased by spurious hits.

The authors' conclusions appear overstated given the limitations in data processing, incomplete methodology, and lack of clear statistical effect size reporting.

We have made a concerted effort to address each limitation highlighted by the reviewer. More specifically, we have significantly expanded the Methods section to provide clear details regarding library preparation, RNA sequencing and bioinformatics processing of the data. We have also added effect size details throughout the manuscript and abstract.

F. Suggested Improvements

I would suggest to provide a detailed description of library preparation, rRNA depletion, host decontamination strategy, and sequencing protocol (read length, single vs paired-end) as well as justify the 2 million read subsampling threshold or increase read depth to improve microbiome resolution.

We appreciate these suggestions and have now significantly expanded the Methods section of the manuscript as follows:

Line 378: Mid-turbinate nasal swabs were collected within 72 hours of hospital admission and at subsequent visits with target dates of 4, 7-, 14-, 21-, and 28-days post hospital admission. The nasal swabs were stored in 1 mL of Zymo-DNA/RNA shield reagent (Zymo

Research), before RNA was extracted twice in parallel from 250 uL of sample. The RNA was then purified with the KingFisher Flex sample purification system (ThermoFisher) and the quick DNA-RNA MagBead kit (Zymo Research).

Extracted RNA then underwent rRNA depletion, cDNA synthesis, and library construction using the Illumina Total Stranded RNA Prep with Ribo-Zero Plus kit, following the manufacturer's instructions, and automated on a Perkin Elmer Sciclone NGSx Workstation. Control samples (Thermo Fisher, cat# AM7832) were also included to assess for and minimize inter-batch variability. Prior to sequencing, libraries were quantified using the Quant-it dsDNA High Sensitivity assay (Invitrogen), fragment size profiles were assessed using an Agilent fragment analyzer, and any samples with > 4% adapter dimers were removed. Libraries, normalized to 10 nM, underwent paired-end 100 base pair sequencing on an Illumina NovaSeq 6000 instrument using S4 flow cells, targeting 50 million reads per sample.

And:

Line 410: Metatranscriptomic data were processed using the open-source CZ ID (<https://czid.org/>) Illumina mNGS pipeline (Nasal Swab data: v7.1)^{40,41}. Host reads were first removed by subtractive alignment against the GRCh38 human reference genome⁴² using STAR⁴³. Adapters were then removed using Trimmomatic⁴⁴, low quality reads filtered using PriceSeq⁴⁵ and low complexity reads filtered using the Lempel-Ziv-Welch algorithm. Duplicate reads were identified and compressed using czid-dedup⁴⁰. A final scrub of any remaining host reads was carried out by re-alignment to GRCh38 using Bowtie2⁴⁶. After performing these filtering steps, the remaining microbial data were subsampled to 2 million total reads. Taxonomic classification was then carried out by aligning reads to the NCBI nucleotide (NT) database using Minimap2. In parallel, short reads were assembled into contiguous sequences using (SPAdes)⁴⁹, which then underwent alignment against accessions from the identified taxa to improve mapping specificity^{40,49}.

As discussed above, subsampling occurred after all quality filtering and host sequence removal is complete. Given that respiratory samples are comprised primarily of human sequence, most samples (2513/2846, 88.3%) did not require subsampling at all. The subsampling to 2 million paired-end microbial reads is a function of the CZID pipeline, and was originally incorporated to make comparison across samples more appropriate when high variability exists in the sequencing depth, and to increase processing speed for samples with disproportionately high number of microbial reads. And as noted above, we believe that this subsampling approach is appropriate in our study for two reasons. First, the reads from 88.3% of samples were completely captured by the 2 million read cutoff; thus, we are fully profiling the data for most of the samples. Second, for the remaining samples that were subsampled down, the total number of non-host reads passing QC was highly variable, ranging from just over 2 million to 32 million reads. The subsampling approach implemented in our study allowed us to analyze all data for the majority of samples while rarefying the data for the 11.7% of exceedingly deeply sequenced samples, allowing for more statistically sound comparisons across all samples.

The authors should incorporate higher-resolution taxonomic and functional profiling, potentially using marker-based tools like MetaPhlan instead of CZ ID. They should also display the species and ARGs detected (especially MLS genes) in main figure or supplementary material. This result in itself is crucial as the RNA-based approach may show differences to the structure observed via metagenomics. Furthermore, a representation of the dominant taxa will show the impact of the contamination, if any and also strengthen the quality of the data.

The bioinformatics approach we utilized produces species-level taxonomic assignments; however we chose to perform our primary analyses at the genus level because of superior accuracy of mapping reads at the genus level. That said, among reads mapping at the species level, many are species-specific but inevitably a subset of reads are shared across multiple species within a given genus, and thus may be stochastically assigned to closely related species. While trivial when assessing overall community composition of a microbiome, non-specific mappings to rare but related species may become disproportionately apparent during differential abundance analyses. Regardless, such instances are relatively uncommon and thus we have provided the following additional analyses at the species level:

1. Performing species level multidimensional correlation analyses, similar to the genus-level analyses described in Fig. 4b. We have included results in a new Supplementary Table 4 and have highlighted those species within significant differentially abundant taxa in a new Supplementary Fig. 7.

Supplementary Figure 7. Species-level correlations within the airway resistome and microbiome. Species-level Spearman correlation analysis among differentially abundant genera from Fig. 2f. Only taxa with significant correlations are displayed. Color bar reflects Spearman correlation coefficient. *** $p_{adj} < 0.001$; ** $p_{adj} < 0.01$; * $p_{adj} < 0.05$.

- Employing the Comprehensive Antibiotic Resistance Gene Database pathogen-of-origin tool²⁹ to match k-mers from detected MLS resistance gene sequences against species present in the tool's reference database, which consists of 437 bacterial genomes, most of which represent established pathogens.

Supplementary Figure 8. Species-level taxonomic assignment of MLS resistance genes based on CARD pathogen-of-origin assignments. Heatmap of CARD pathogen-of-origin assignments and MLS ARGs. Red cells denote the assignment of a given ARG to a given taxon by CARD. Genomic location information in parentheses.

While we agree with the reviewer that MetaPhlAn is an excellent tool for stringently and confidently assigning taxonomy from metagenomic and metatranscriptomic data, the tool was designed for profiling richer microbial communities (e.g., stool) with much greater sequencing coverage of the microbiome. In contrast, respiratory samples are mostly comprised of human sequence (average of 99.90%, median of 99.97%, range of 99.00-99.99% human in our study), challenging the utility of MetaPhlAn for comprehensively profiling microbial communities from the respiratory tract, with the exception of accurately calling highly abundant species. While CZ ID assigns taxonomic calls based on short read and contig sequence similarity alone, MetaPhlAn requires that a given taxon has at least 10% of the marker genes from that taxon detected in the data. While strain-level taxonomic identification using MetaPhlAn is generally excellent for the most abundant taxa within the airway microbiome, less abundant but nonetheless important members of the community may be excluded. As such, we believe that our use of the CZ ID pipeline, which has been employed for microbial profiling in hundreds of manuscripts including in *Nature Microbiology* and other Nature family journals, is both rigorous and well suited for the analyses in this manuscript. For example:

- <https://www.nature.com/articles/s41564-022-01237-2>
- <https://www.nature.com/articles/s41467-022-29353-x>
- <https://www.nature.com/articles/s41591-024-03130-3>
- <https://www.nature.com/articles/s41591-024-03274-2>
- <https://www.nature.com/articles/s41559-025-02711-w>
- <https://www.nature.com/articles/s41467-025-55823-z>

We appreciate the reviewer's suggestion to carry out a microbiome functional analysis of the metatranscriptomic data. We have now done so using HUMAnN3. For this analysis, we compared individuals who received 5 +/-1 days of Azithromycin treatment (N=116) to those who received either 5 +/-1 days of other antibiotics or no antibiotics (N=342). Due to the stringency of MetaPhlan4 in assigning taxonomy and the sparsity of upper respiratory samples, only 159/458 (34%) of the samples yielded a MetaPhlan4 taxonomic profile, necessary for HUMAnN3 functional profiling. Despite these limitations, we analyzed the data using both a 1% and 5% pathway prevalence cutoff (202 vs 96 pathways) and found 0 significant pathways in either analysis after adjusting for multiple comparisons. We have added the following to the manuscript to address this:

Line 163: We next performed a functional analysis of microbial metabolic pathway expression using HUMAnN3²⁷ and MaAsLin2²⁸ to compare the Azithro group to the Other-Abx and No-Abx groups at the 5 ±1 day timepoint, but found zero differentially abundant pathways after adjusting for multiple comparisons. We also tested whether azithromycin treatment associated with any changes in SARS-CoV-2 relative abundance in the upper airway. We observed no differences with respect to either the Other-Abx or No-Abx groups after 5 ±1 days of treatment (Supplementary Fig. 3).

Methods, Line 537: Microbiome data were functionally annotated using HUMAnN 3²⁷ and renormalized using the `humann_renorm_table` function. MaAsLin2²⁸ was used to identify differentially abundant pathways in the normalized relative abundance data, with a prevalence cutoff of 1%, a minimum_abundance cutoff of zero and log transformation.

G. References: Appropriate Credit to Previous Work?

Some established associations between taxa and ARGs are missing. For instance, the known linkage between *Prevotella* and *ermB* (e.g., <https://doi.org/10.3390/pathogens10121546>) is not acknowledged.

We appreciate the reviewer highlighting this study which we have now cited in the manuscript and in the source data (literature references) for Figure 4c.

H. Clarity and Context

Another minor issue is the terminology used in the manuscript. The term bacterial abundance used by the authors is based on the reads per millions which is ambiguous because it is still a relative abundance rather than a true quantitative abundance.

We understand the reviewer's point and have incorporated their suggestion by changing abundance to *relative abundance* throughout the manuscript.

Reviewer #1 (Remarks to the Author):

The authors have responded to my comments and it seems that those are all well addressed.

We appreciate the reviewer's thoughtful feedback and constructive suggestions, which greatly strengthened our work. We thank them for their time and effort throughout the review process.

Reviewer #2 (Remarks to the Author):

The authors have addressed all my initial concerns and suggestions. No further changes from my end.

We are grateful for the reviewer's evaluation and valuable input, which helped to refine our manuscript. We thank them for their time and effort throughout the review process.

Reviewer #3 (Remarks to the Author):

The authors have answered most of my comments except for one major point and one minor point.

The major point concern the rationale behind limiting analysis to 2 million reads per sample which is still, to my point of view, not justified or at least limit the conclusion because 2 Mio reads, even after cleaning, is quite low to explore the full diversity of the microbiome. I would appreciate it if the authors could provide a rarefaction curve showing that with 2 Mio reads, they are reaching the plateau. My argument is also strengthened by the fact that the Metaphlan provided showed that only 159/458 (34%) of the samples yielded a MetaPhlan4 taxonomic profile, which then indicates that the genome coverage is also quite low.

We appreciate the opportunity to further address microbial read subsampling. We would first like to note that microbial (non-host) reads represented a very small fraction of total reads per sample, as expected for respiratory samples. As such, only 11.7% of samples ultimately underwent subsampling to 2 million reads.

Regardless, as requested, we have now generated rarefaction curves for both antimicrobial resistance genes (ARGs) and microbial taxa. Both demonstrate a plateau in richness prior to 2 million microbial reads.

ARG richness vs ARG reads, individual patient level:

ARG richness vs ARG reads, composite:

Genus richness vs microbial reads, individual patient level:

Genus richness vs microbial reads, composite:

Species richness vs microbial reads, individual patient level:

Species richness vs microbial reads, composite:

The minor comment concerns the PERMANOVA analysis.

The author claims that PERMANOVA does not give an R^2 or effect size value. However, PERMANOVA calculates a pseudo-F statistic to test for differences between groups, which then represents the ratio of the between-group sum of squares to the within-group sum of squares (or mean squares). Furthermore, several PERMANOVA test, such as the ADONIS test, provides an R^2 value that measures how much of the total variation in the data is explained by the grouping factor.

We thank the reviewer for correcting us on this point. We used the ADONIS test and have now included the R^2 statistics for each PERMANOVA analysis in the figure legends. In addition, we have provided the R^2 and F statistics for each in the Source Data file.

Microbial PERMANOVA

	Df	Sum of Sqs	R^2	F	Pr(>F)
Day 1 \pm 1 vs No ABX	1	4.45	0.00922	9.688	0.000999
Residual	1027	471.63	0.97782		
Total	1036	482.33	1		
	Df	Sum of Sqs	R^2	F	Pr(>F)
Day 5 \pm 1 vs No ABX	1	3.179	0.01703	7.1779	0.000999
Residual	405	179.353	0.96079		
Total	414	186.673	1		
	Df	Sum of Sqs	R^2	F	Pr(>F)
Day 1 \pm 1 vs Day 5 \pm 1	1	0.74	0.0019	1.5686	0.018981
Residual	812	384.4	0.98194		
Total	821	391.47	1		
	Df	Sum of Sqs	R^2	F	Pr(>F)
Three-way comparison	2	5.75	0.01082	6.2361	0.000999
Residual	1126	519.54	0.97695		
Total	1136	531.8	1		

AMR GENE PERMANOVA

	Df	Sum of Sqs	R ²	F	Pr(>F)
Day 1 ± 1 vs No ABX	403	182.626	0.71658	1.3489	0.000999
Residual	215	72.231	0.28342		
Total	618	254.857	1		
	Df	Sum of Sqs	R ²	F	Pr(>F)
Day 5 ± 1 vs No ABX	205	89.693	0.84536	1.4666	0.000999
Residual	55	16.408	0.15464		
Total	260	106.101	1		
	Df	Sum of Sqs	R ²	F	Pr(>F)
Day 1 ± 1 vs Day 5 ± 1	261	120.63	0.61985	1.3744	0.000999
Residual	220	73.98	0.38015		
Total	481	194.61	1		
	Df	Sum of Sqs	R ²	F	Pr(>F)
3-way comparison	424	194.152	0.69518	1.377	0.000999
Residual	256	85.129	0.30482		
Total	680	279.282	1